# Global profiling of distinct cysteine redox forms reveals wide-ranging redox regulation in *C. elegans*

Jin Meng [1,2,3,8], Ling Fu[4,5,8], Keke Liu[4], Caiping Tian[4,6], Ziyun Wu[1,2,3], Youngeun Jung[7], Renan B. Ferreira [7], Kate S. Carroll [7], T. Keith Blackwell[1,2,3✉] & Jing Yang [4,5✉]

Post-translational changes in the redox state of cysteine residues can rapidly and reversibly alter protein functions, thereby modulating biological processes. The nematode *C. elegans* is an ideal model organism for studying cysteine-mediated redox signaling at a network level. Here we present a comprehensive, quantitative, and site-specific profile of the intrinsic reactivity of the cysteinome in wild-type *C. elegans*. We also describe a global characterization of the *C. elegans* redoxome in which we measured changes in three major cysteine redox forms after $H_2O_2$ treatment. Our data revealed redox-sensitive events in translation, growth signaling, and stress response pathways, and identified redox-regulated cysteines that are important for signaling through the p38 MAP kinase (MAPK) pathway. Our in-depth proteomic dataset provides a molecular basis for understanding redox signaling in vivo, and will serve as a valuable and rich resource for the field of redox biology.

[1] Research Division, Joslin Diabetes Center, Boston, MA, USA. [2] Department of Genetics, Harvard Medical School, Boston, MA, USA. [3] Harvard Stem Cell Institute, Cambridge, MA, USA. [4] State Key Laboratory of Proteomics, Beijing Proteome Research Center, National Center for Protein Sciences (Beijing), Beijing Institute of Lifeomics, Beijing, China. [5] Innovation Institute of Medical School, Medical College, Qingdao University, Qingdao, China. [6] School of Medicine, Tsinghua University, Beijing, China. [7] Department of Chemistry, The Scripps Research Institute, Jupiter, FL, USA. [8]These authors contributed equally: Jin Meng, Ling Fu. ✉email: keith.blackwell@joslin.harvard.edu; yangjing54@hotmail.com

Reactive oxygen species (ROS) are constantly generated within mitochondria, the endoplasmic reticulum (ER), peroxisomes, and at cellular membranes[1]. Those highly reactive molecules play a dual role in eukaryotes: excessive ROS induce oxidative stress and, therefore, contribute to human diseases, while low levels of ROS act as signaling molecules and regulate biological processes physiologically[2,3]. ROS production is affected by various stimuli, such as growth factors and cytokines[1]. For example, epidermal growth factor (EGF) or insulin induces the production of hydrogen peroxide ($H_2O_2$) by NADPH oxidases, which promotes the activation of those signaling pathways by oxidizing downstream effectors, including protein tyrosine phosphatases (PTPs)[4].

Accumulating evidence suggests that ROS signaling is involved in a broad range of biological processes, such as metabolism, aging, and oxidative stress defense[2,3,5–10]. The most prominent mode of ROS signaling involves their reaction with the nucleophilic thiol group (–SH, reduced form) in specific protein cysteines, which results in an array of oxidative post-translational modifications (oxiPTM). In the presence of ROS, thiols are initially oxidized into S-sulfenic acids (–SOH), which are highly reactive and can be converted into more stable forms, including S-sulfinic acids (–SO2H) and disulfides (–SS–) (Fig. 1)[1]. These reversible, regulatory oxiPTMs on specific cysteines have emerged as important mechanisms that alter protein function post-translationally[1,2,6,10–12].

The collection of redox-regulated proteins comprises the redoxome. Advances in proteomic techniques have made it possible to quantify in different species the extent to which specific cysteines are oxidized reversibly[7,13–15], work that has greatly expanded understanding of the redoxome. Studies that measured absolute stoichiometric cysteine oxidations have found that perhaps most thiols become oxidized to only a small extent under physiological conditions[7,13–15]. However, increasing the percent modification of a specific cysteine in the kinase AKT from 0.5% to only 1.25% has been found to have physiological consequences[16], indicating the importance of comprehensively identifying cysteines that are prone to modification. To understand redox regulation, it is also important to profile specific redox forms,

because distinct oxidative modifications can have different biological impacts[17]. Compared to the advances in the mapping of reversible thiol oxidations, specific oxidized forms of cysteines have been historically difficult to analyze biochemically. With the recent development of specific probes for S-sulfenic acids or S-sulfinic acids, hundreds of peroxide-induced cysteine oxidation events have been identified in plant and human cells[4,18,19]. The large number of probable redox-sensitive cysteines across the proteomes suggests that diverse physiological processes may be subject to redox regulation, and highlight the need for biochemical and genetic assays to elucidate functional significance of individual cysteine residues in vivo.

Because of its genetic tractability, the nematode C. elegans provides a particularly advantageous model for studying roles for functionally important cysteines in redox signaling in vivo. For instance, peroxide-induced formation of intramolecular disulfide bonds results in the inactivation of the histone methyltransferase SET-1/MLL1, and such redox regulation is conserved between worms and mammals[6]. Importantly, the redox status of a single cysteine can redirect biological processes in response to changes in cellular redox homeostasis. One example of such redox switches is C663 within the kinase activation loop of the ER protein IRE-1[10]. ROS-induced S-sulfenylation on C663 suppresses the unfolded protein response (UPR), but simultaneously stimulates an antioxidant response mediated by the p38 mitogen-activated protein kinase (MAPK) signaling[10]. Despite those studies that focused on individual cysteines, examinations of the C. elegans redoxome to date have covered a relatively small proportion of the cysteine proteome (cysteinome), and analyses of different cysteine redox forms have been lacking. Pioneering work by Jakob and coworkers quantified the thiol redox states in more than one hundred proteins in C. elegans[14,20]. More recently, Weerapana and coworkers measured cysteine reactivity in mutant worms with altered lifespan, and identified 50 cysteines that were affected by insulin/insulin-like growth factor-1 (IGF-1) signaling[21]. Those studies displayed the feasibility of conducting redox proteomics in C. elegans, and underscored the power of such analyses in understanding how ROS control physiological signaling events in this organism.

Taking advantage of recent advances in state-of-the-art site-centric chemoproteomic approaches and the synthesis of proteomics-compatible probes, here we have comprehensively determined the intrinsic reactivity of the C. elegans cysteine proteome under physiological conditions, and globally assessed changes in three types of redox forms (–SH, –SOH, and –SO2H) that occur rapidly after a brief treatment with a modest dose of $H_2O_2$. This effort identifies thousands of cysteine redox forms that are relatively susceptible to oxidation, and thus represent potential sites of redox regulation. We find that many proteins involved in translation, histone modification, pre-mRNA splicing, and growth regulatory pathways are enriched in the peroxide-dependent redoxome, suggesting that those processes are modulated by redox signaling. Illustrating the value of this database, we identify previously unknown ROS targeted cysteines in the p38 MAPK signaling pathway, experimentally substantiate the importance of these cysteines in oxidative stress response and innate immunity, and indicate that redox regulation occurs at each step of this pathway. Together, we have systematically and quantitatively defined the C. elegans redoxome with deep coverage, identified many biological processes and pathways in the redox signaling network, and demonstrated molecular mechanisms underlying the redox regulation of the p38 signaling pathway.

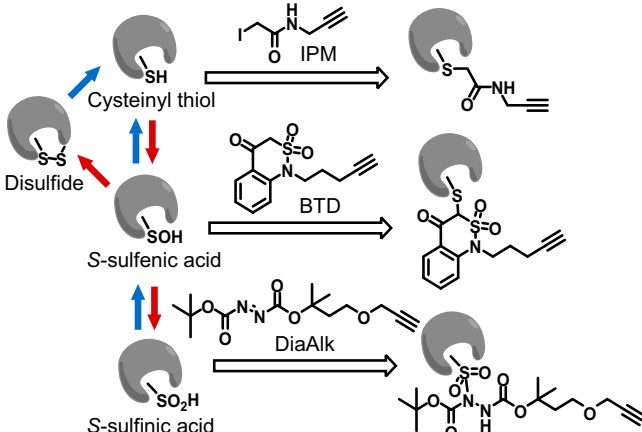

**Fig. 1 Labeling of cysteine redox forms with chemoselective probes.** IPM is an iodoacetamide-based alkyne probe for cysteinyl thiol (–SH)[66]; BTD is a benzothiazine-based alkyne probe for cysteine sulfenic acid (–SOH)[29]; DiaAlk is a diazene-based alkyne probe for cysteine sulfinic acid (–SO2H)[19]. The alkyne group can be conjugated to azide-bearing tags via click chemistry for detection and enrichment. In mammalian cells, –SO2H within peroxiredoxins can be reduced by a sulfiredoxin[67]. In C. elegans no known sulfiredoxin is present, and –SO2H is presumably irreversible[33]. Oxidation and reduction processes are depicted in red and blue arrows, respectively.

## Results

**Mapping hyperreactive cysteines in C. elegans.** Despite their relatively low abundance in proteins, cysteine residues are often found at functional sites and may participate in processes, such as

catalysis, ligand binding, and protein–protein interaction[17]. Those functional cysteines generally display high intrinsic reactivity, and thus it has been proposed that cysteine reactivity could predict functionality[21,22]. To quantify cysteine reactivity on a proteome-wide scale, in a pioneering work, Cravatt and coworkers employed a dose-dependent labeling of native proteomes with a "clickable" thiol-reactive probe[22]. In principle, hyperreactive cysteines would saturate labeling at the low probe concentration, whereas less labile cysteines would show concentration-dependent increases in labeling. Application of this strategy in mammalian cells reveals that heightened reactivity indeed serves as a good predictor of cysteine functionality[22], even though some intrinsically reactive and functional cysteines might be inaccessible to the probe due to chemical or steric reasons. Based on the same principle, we recently developed a chemoproteomic method named quantitative thiol reactivity profiling (QTRP), and systematically profiled cysteine reactivity in *Drosophila* and mammalian cells[23,24].

In this study, we first globally profiled cysteine reactivity in *C. elegans* by applying QTRP to wild-type L4 stage animals. Worms were lysed under non-denaturing conditions, and then treated with either $100\,\mu M$ or $10\,\mu M$ of the IPM probe (Fig. 2a). The ratio of labeling by high and low doses of IPM ($R_{100:10}$) was determined for a total of 5258 cysteines, with 52.0% (2735) quantified in at least two of the three biological replicates (Supplementary Data 1). Notably, >90% of the reproducibly quantified sites showed a coefficient of variation (CV) value lower than 40% with a medium value of 13.4%, demonstrating the reproducibility of our data (Supplementary Fig. 1). By mapping approximately 4400 additional cysteine sites, our results substantially expanded the coverage of the *C. elegans* cysteinome obtained in a previous study of insulin-like signaling pathway mutants[21] (Fig. 2b). 24.1% (107 of 444) of all cysteine sites that were detected in both studies exhibited similar $R_{100:10}$ values, regardless of a difference in genetic backgrounds (wild-type or *daf-16; daf-2* double mutants) (Fig. 2c).

We defined cysteines with $R_{100:10}$ values below or equal to 3.0 as hyperreactive, and those with $R_{100:10}$ values higher than 3.0 but not higher than 6.0 as moderately reactive. The cutoff ratio values were defined empirically based on previously published reports[21,22,24] for the sake of simplicity and consistency. Overall, hyperreactive cysteines accounted for 24.6% (1292 out of 5258) of all quantified sites, while moderate ones represented 43.1% (2266 out of 5258). Because of incomplete functional annotation information in the UniProt knowledge base, it was challenging to locate all reactive cysteines within specific protein domains and thereby assess their predicted functional importance (Fig. 2d). Despite that, many known catalytic and/or redox-sensitive cysteines were highly reactive. For example, the evolutionarily-conserved active site C36[25] in glutathione peroxidase 2 (GPX-2) showed an $R_{100:10}$ value of 2.68 (Supplementary Data 1). We noticed that some non-catalytic cysteines were more reactive than known active or redox-sensitive sites within the protein. For instance, the active cysteine C681[26] in the ubiquitin-activating enzyme UBA-1 showed a ratio of $R_{100:10} = 5.49$, while the non-catalytic residue C543 was more reactive with an $R_{100:10}$ of 2.32 (Fig. 2e). One possibility is that reactive cysteines might function in a non-catalytic manner, or might be modified through other routes than oxidation (e.g., S-palmitoylation, lipid electrophile-based S-adductions).

Our cysteine reactivity dataset included many conserved cysteines for which reactivity was assessed previously in cultured human cells[22]. Interestingly, 54% of these cysteines were detected with similar intrinsic reactivity in both species (Supplementary Fig. 2). For example, the active cysteine C33 in the glutathione S-transferase GSTO-1 exhibited almost the same reactivity in

*C. elegans* ($R_{100:10} = 1.1$) (Fig. 2f) and *H. sapiens* ($R_{100:10} = 0.9$)[22]. Notably, conserved cysteines within paralogs often, but not always, display similar $R_{100:10}$ values. For instance, C158 is an active site within all four orthologs (GPD-1 through GPD-4) of human glyceraldehyde 3-phosphate dehydrogenase (GAPDH)[27]. C158 in GPD-1 and GPD-2/3 showed very similar $R_{100:10}$ values of 5.89 and 5.27, though C158 of GPD-4 exhibited a higher $R_{100:10}$ of 10.0 (Fig. 2g). Those results suggest that conservation at the level of amino acid sequence may generally correlate with cysteine intrinsic reactivity, although reactivity could be influenced by subtle changes in flanking sequences.

In conclusion, by identifying 3558 intrinsically reactive cysteines ($R_{100:10} \le 6$), our findings greatly expanded the landscape of the reactive cysteine proteome in *C. elegans*. Our analysis revealed that dramatic differences in reactivity may exist for different cysteines within the same protein, and that conservation of cysteine residues to an extent predicts their redox-reactivity and possible redox regulation of their functionality.

**Proteome-wide profiling of oxidant-sensitive cysteine redox forms in *C. elegans*.** The reactivity of a cysteine is determined by its reaction kinetics with the electrophilic probe IPM, which may not correspond exactly with its interactions with ROS (e.g., $H_2O_2$). As a result, the intrinsic reactivity may predict the functionality of a cysteine within a broad range but does not directly show its potential for redox regulation. Therefore, we next sought to directly assess redox sensitivity of cysteines in *C. elegans* at a proteome-wide scale.

We recently developed site-centric chemoproteomic methods that can precisely and quantitatively measure major regulatory cysteine redox forms in complex proteomes[19,28,29]. Specifically, the chemoselective probes IPM, BTD, and DiaAlk allow labeling of –SH, –SOH, and $SO_2H$, respectively (Fig. 1). To define the *C. elegans* redoxome with respect to these specific cysteine forms, we adopted these established methods for each probe in wild-type worms[19,28,29] and evaluated changes in response to a 5-min treatment of $5\,mM$ $H_2O_2$ (Fig. 3a). Such a treatment avoids significant perturbations to protein levels, but also allows the detection of early changes in cysteine oxidation and the most oxidation-sensitive cysteine residues, without affecting the animal's viability or behaviors[30,31]. For each cysteine detected in these analyses, we calculated a treated/control ratio ($R_{T/C}$). Oxidation of a thiol by $H_2O_2$ would reduce its accessibility to the thiol-reactive probe IPM, and thus a lower $R_{T/C}^{IPM}$ indicates increased cysteine oxidation. Meanwhile, $H_2O_2$-induced formation of S-sulfenic acid or S-sulfinic acid would enable more BTD or DiaAlk-derived chemoselective conjugation, thereby rendering relatively high $R_{T/C}^{BTD}$ or $R_{T/C}^{DiaAlk}$ values. In total, we mapped and quantified 5453 Cys-SH sites on 2864 proteins, 1521 Cys-SOH sites on 1049 proteins, and 82 Cys-$SO_2H$ sites on 72 proteins (Fig. 3b, Supplementary Data 2, 3, and 4).

Because cysteine oxidation is transient and reversible, relatively large quantitative variations among samples are often observed[16]. However, notably, 3301 cysteine-mediated redox events were quantified in more than one replicate, with medium CV values ranging from 8.4% to 20.4% (Supplementary Fig. 1), demonstrating the reproducibility of our analyses. Our dataset included numerous cysteines that are known to be redox regulated and/or have been functionally annotated in the UniProt knowledge base. Due to the stochastic nature of shotgun proteomics, it was not surprising that many cysteine redox forms were detected in only one replicate, also including known active or redox-sensitive cysteines. One of such example was a *bona fide* redox-sensitive cysteine (C52, $R_{T/C}^{IPM} = 0.46$, Supplementary Data 2) in the CXXC motif of protein disulfide-isomerase 1 (PDI-1). This result

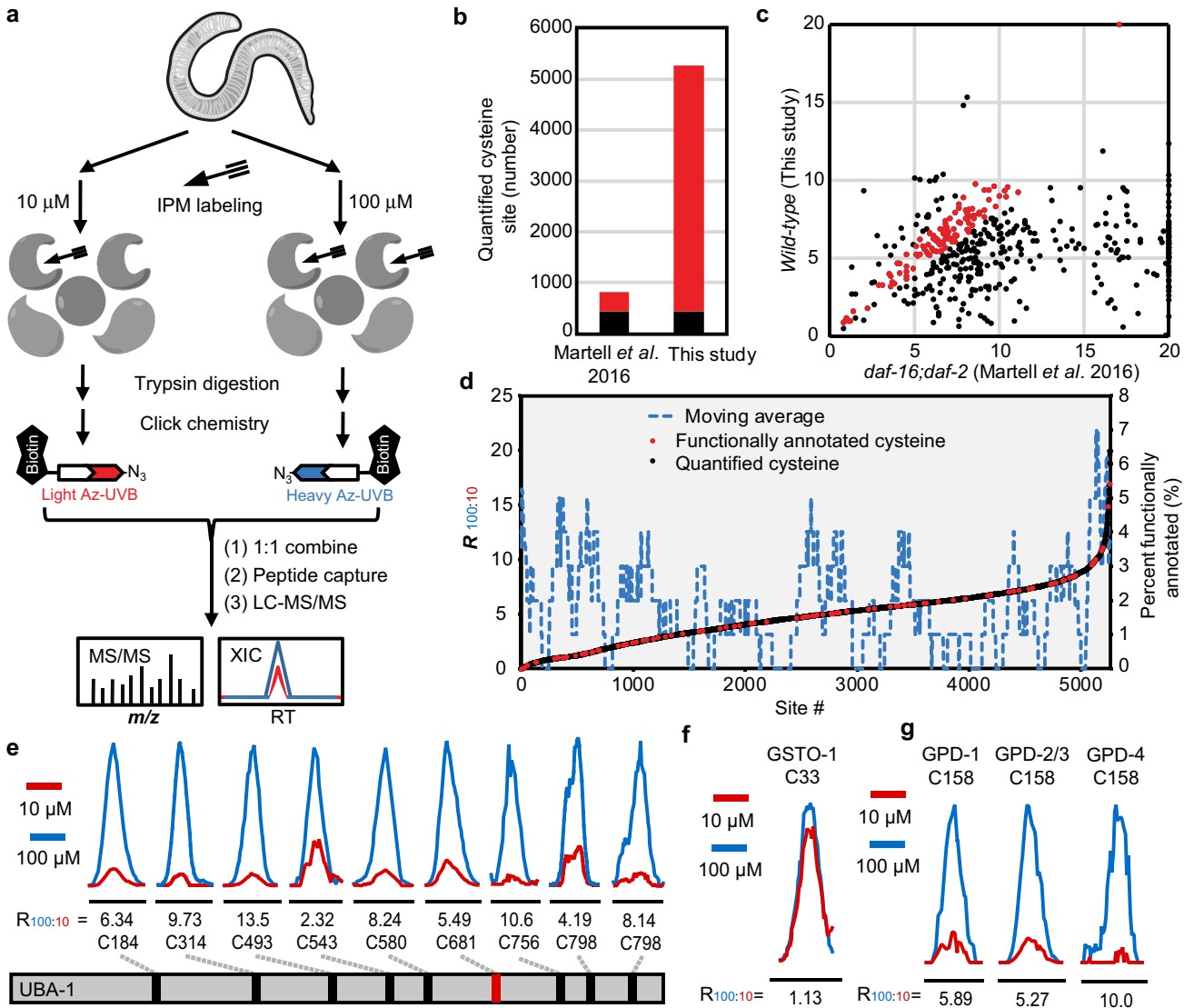

**Fig. 2 Mapping hyperreactive cysteines in *C. elegans*. a** Schematic diagram of our quantitative chemoproteomic workflow for site-specific quantification of the intrinsic reactivity of cysteines in the *C. elegans* proteome. Lysates of *C. elegans* harvested under the same condition were labeled with either 10 or 100 μM IPM, respectively, and digested by trypsin. The resulting IPM-modified peptides were conjugated to light (10 μM, in red) or heavy (100 μM, in blue) azido biotin reagents with a photocleavable linker (Az-UV-biotin) via CuAAC, also known as click chemistry. The light and heavy-labeled samples were then mixed equally in amount and subjected to streptavidin-based enrichment. After several washing steps, the modified peptides were selectively eluted from beads under 365 nm wavelength of UV light for LC-MS/MS-based proteomic analysis. **b** Bar chart showing the numbers of quantified cysteine sites in two different studies, with common sites in black and different sites in red. **c** Scatter plot showing the $R_{100:10}$ values measured for cysteines quantified in both studies, and those with similar $R_{100:10}$ values (less than 1.5-fold difference) in both studies are colored in red. **d** Correlation of $R_{100:10}$ values with functional annotations from the UniProt database, where active sites, disulfide bonds, or metal-binding sites are shown in red, and all other quantified cysteines are in black. A moving average line of functional annotated sites is shown in a dashed blue line. **e–g** Representative extracted ion chromatograms (XICs) showing changes in IPM-labeled peptides from UBA-1 (**e**, active site is shown in red), GSTO-1 (**f**), and GPDs (**g**). The profiles for light- and heavy-labeled peptides are shown in red and blue, respectively. The average $R_{100:10}$ values calculated from biological triplicates are displayed below each XIC.

suggests that our dataset serves as a useful resource for discovering functionally important redox events in *C. elegans*, and that even "single hits" may be physiologically relevant.

Because sulfenylation is the first step towards sulfinylation (Fig. 1)[1], it was not surprising that approximately 78% of S-sulfinylated sites were identified as S-sulfenylated (Supplementary data 3 and 4). Although it has been suggested that oxidation of a thiol to S-sulfenic acid occurs more readily than the conversion of S-sulfenic into S-sulfinic acid[32], we found that acute $H_2O_2$ treatment caused greater changes in S-sulfinylation than in S-sulfenylation on many cysteines (Fig. 3c). For example, the largest increase in peroxide-induced sulfinylation occurred on C120 in

the protein Y41D4A.5, an ortholog of mammalian protein tyrosine phosphatase non-receptor type 22 (PTPN22), with an $R_{T/C}^{DiaAlk}$ of 7.01, while both the –SH and –SOH forms of this cysteine altered moderately ($R_{T/C}^{IPM} = 0.58$ and $R_{T/C}^{BTD} = 1.57$) (Fig. 3d). As a second example, the –SO₂H form of the active site C158 in GPD-3 exhibited great sensitivity to $H_2O_2$ treatment ($R_{T/C}^{DiaAlk} = 4.45$), whereas neither the –SH nor –SOH form changed much ($R_{T/C}^{IPM} = 0.52$ and $R_{T/C}^{BTD} = 1.35$) (Fig. 3e). One possible reason for the dramatic increase in C158 S-sulfinylation level could be its inability to form a disulfide bond, due to the long distance of ~8.8 Å between Sγ(158) and the closest sulfur Sγ(162) (Fig. 3f). In general, the wide dynamic range

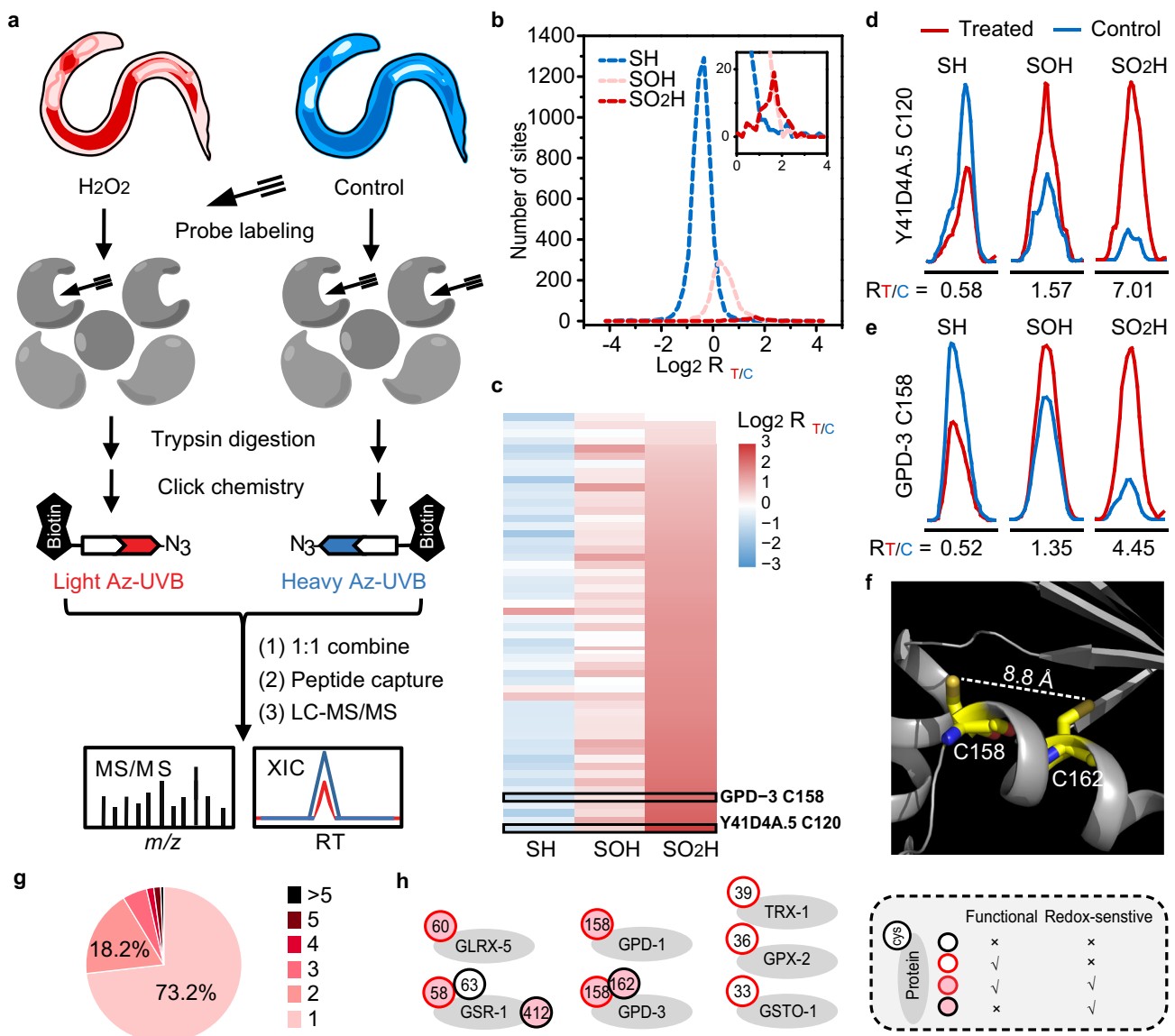

**Fig. 3 Defining the oxidation-sensitive *C. elegans* redoxome. a** Schematic diagram of our quantitative chemoproteomic workflow for profiling cysteine redox forms in *C. elegans* upon peroxide treatment. Worms treated with (red) or without (blue) $H_2O_2$ were labeled in vitro with the chemoselective probes IPM (for –SH), BTD (for –SOH), or DiaAlk (for –$SO_2H$), in parallel (Fig. 1). The probe-tagged proteomes were processed into tryptic peptides, followed by reactions with light or heavy Az-UV-biotin reagents via CuAAC. The light and heavy-labeled samples were then mixed equally in amount, cleaned with SCX, and enriched on streptavidin beads. Labeled peptides were then photoreleased and subjected to LC-MS/MS-based proteomic analysis for identification and quantification of individual cysteine residues. **b** Distribution of the Log-transformed $R_{T/C}$ values for cysteine redox forms. **c** Heatmap showing dynamic changes in redox forms for cysteine sites identified by all three probes. **d**, **e** Representative XICs showing changes in probe-labeled peptides from Y41D4A.5 (C120) (**d**) and GPD-3 (C158) (**e**). Profiles for light- and heavy-labeled peptides are shown in red and blue, respectively. The average $R_{T/C}$ values calculated from two biological replicates are displayed below each XIC. **f** Structure of GPD-3 ortholog showing the distance between C158 and C162. **g** A Pie chart showing the distribution of the number of dynamically-changed cysteine redox forms per protein in the *C. elegans* redoxome. **h** A selected group of known redox-reactive metabolic enzymes identified in this study. Functionality information is retrieved from the UniProt knowledge base.

of S-sulfinylation could be partially attributed to a lack of an efficient reduction system for S-sulfinic acids in *C. elegans*, which makes cysteines highly susceptible to overoxidation[33]. Moreover, the large number of sulfenylated proteins suggests that basal sulfenylation levels might be high in *C. elegans*, which also allows overoxidation into S-sulfinic acids to occur more readily.

Using a common criterion in redox proteomics ($R_{T/C}^{IPM} \leq$ 0.67, $R_{T/C}^{BTD} \geq 1.5$, or $R_{T/C}^{DiaAlk} \geq 1.5$)[18,34], 33.2% (1813/5453), 31.0% (472/1521), and 93.9% (77/82) of –SH, –SOH, and –$SO_2H$ sites in a total of 1537 proteins changed dramatically upon peroxide treatment (Fig. 3g and Supplementary Data 5).

We consider these sites as comprising the peroxide-sensitive *C. elegans* redoxome. This large redoxome dataset included a number of known redox-sensitive cysteines, such as C158 in GPD-1[27] (Fig. 3h), C35 in the 40S ribosomal protein RPS-17[20] (Supplementary Data 3), and C663 in the AGC family kinase IRE-1[10] (Supplementary Data 2). However, we noticed that our database did not include the peroxidatic (or catalytic) cysteines in 2-Cys peroxiredoxins (e.g., PRDX-2) that are well known to be redox-reactive[30]. To maintain the intrinsic reactivity of profiled cysteines, we employed native lysis conditions[23], under which the peroxidatic cysteine in PRDX-2 was predominantly disulfide-

linked and therefore would not have been labeled and detected (Supplementary Fig. 3). The absence of this known redox-regulated cysteine in our dataset may also be attributed to the long and hydrophobic tryptic peptide bearing this site that may have be missed in liquid chromatography-tandem mass spectrometry (LC-MS/MS) detection. In addition, the oxidation state of cysteines is spatially and temporally controlled and could be affected by a protein's subcellular localization and ROS levels. This might explain the lack of peroxide-induced changes in some other known redox-sensitive sites in our dataset (Fig. 3h). Importantly, however, the presence of a very large number of known and newly identified redox-regulated proteins in our dataset made us confident about its coverage and validity.

**Cysteine redox modifications modulate various biological processes and pathways.** We next leveraged our redoxome data to generate profiles of tissues and cellular compartments. As expected, oxidant-modified proteins were expressed in various tissues and organs, such as germ line, muscle, and intestine (Supplementary Fig. 4). Those proteins were widely distributed in major cellular organelles, including mitochondria, the ER, and nucleus (Fig. 4a). Gene ontology (GO) and KEGG enrichment analyses of the *C. elegans* redoxome identified a myriad of essential cellular processes. The ubiquitin–proteasome pathway represented a large group of oxidant-reactive proteins (Fig. 4b). For instance, thiol availability by IPM on the active cysteine C89 of the E2 ubiquitin-conjugating enzyme UBC-14[35] dramatically decreased upon $H_2O_2$ stimulus ($R_{T/C}^{IPM} = 0.36$, Supplementary Data 2). The enrichment of several metabolic processes including glycolysis, the tricarboxylic acid cycle (TCA) cycle, and the pentose phosphate pathway (Fig. 4b and c) was consistent with a model in which cysteine modifications in redox-sensitive metabolic enzymes alter metabolic fluxes to restore cellular redox homeostasis.

Of interest, two pathways that regulate development, growth, and lifespan were significantly enriched in the redoxome by bioinformatic analyses: the mechanistic target of rapamycin complex 1 (mTORC1) and insulin/IGF-1 signaling pathways (Fig. 4b and c). Two core components of the mTORC1 complex, the serine/threonine protein kinase LET-363/mTOR (C848 with an $R_{T/C}^{IPM} = 0.65$, Fig. 4d) and MLST-8 (C263 with an $R_{T/C}^{IPM} = 0.59$, Supplementary Data 2), contained oxidant-sensitive cysteines. Moreover, the GTPase RAGA-1, which functions in amino acid sensing upstream of mTORC1 activation, also harbored a peroxide-reactive cysteine C167 ($R_{T/C}^{IPM} = 0.66$, Supplementary Data 2) in the GTPase domain close to the GTP/magnesium-binding site. The identification of those oxidant-responsive cysteines in the mTORC1 signaling pathway in *C. elegans* together with previous evidence showing ROS affecting mTORC1 signaling in mammalian cells[36,37] suggests that the redox regulation of this pathway might be evolutionarily conserved. Insulin promotes increased production of $H_2O_2$ at cellular membranes[1], and several proteins in the insulin/IGF-1 pathway are subject to cysteine-mediated redox modulation[11,16]. The transcription factor DAF-16/forkhead box O (FOXO) acts as a key downstream effector of insulin[38], and a disulfide bond between DAF-16/FOXO and the nuclear receptor IMB-2/transportin-1 contributes to its nuclear accumulation in the presence of ROS[11]. Interestingly, we detected additional oxidant-sensitive cysteine residues in DAF-16 itself and its upstream regulators in the insulin/IGF-1 signaling pathway, including C433 ($R_{T/C}^{IPM} = 0.65$, Fig. 4e) in DAF-16, C1394 ($R_{T/C}^{IPM} = 0.52$) in DAF-2, and C124 ($R_{T/C}^{IPM} = 0.67$) and C573 ($R_{T/C}^{IPM} = 0.59$) in DAF-18/PTEN (Supplementary Data 2). C1394 of DAF-2 is proximal to the kinase active site D1388 (UniProt), while C124 of DAF-18 is located in the HCXXGXXR motif that is characteristic of the active sites of PTPs[39]. Those oxidant-modified cysteines might also contribute to ROS-induced DAF-16 translocation into the nucleus (Fig. 4f and g, Supplementary Fig. 5a and b).

Many proteins that influence gene expression were highly represented in the redoxome. Histone-modifying enzymes affect transcriptional activity by controlling chromatin accessibility. We found that several histone-modifying enzymes were highly sensitive to oxidation, such as histone deacetylase HDA-1 and histone-lysine N-methyltransferase MES-2 (Supplementary Data 6), suggesting that they might contribute to altered histone modifications upon oxidative stress[6]. Pre-mRNA splicing regulates gene expression post-transcriptionally, and alternative splicing increases the diversity of mRNAs. Within the *C. elegans* redoxome we also found RNA-binding proteins, including ATP-dependent RNA helicases, serine and arginine-rich (SR) proteins, and other splicing factors (Supplementary Data 6), suggesting that mRNA splicing and transcript profiles might be modulated by cellular redox states.

Our GO term analysis also shows that protein synthesis is likely to be broadly influenced by redox signaling (Fig. 4b). As one translation regulatory mechanism, previous work in *C. elegans* and yeast indicates that ROS inhibit protein synthesis by enhancing the general control non-depressible protein 2 (GCN2)-dependent phosphorylation of the α subunit of eukaryotic initiation factor 2 (eIF2α)[40,41]. In yeast, eIF2α phosphorylation by GCN2 under stressed conditions requires the GCN2-interacting partner GCN1[42] (Fig. 4h). Intriguingly, we found that the GCN1 ortholog in *C. elegans* was widely oxidized in response to $H_2O_2$, with 11 oxidant-sensitive cysteines and 4 S-sulfenylated sites mapped (Fig. 4i and Supplementary Fig. 6a). To investigate the potential role for GCN-1 in GCN-2 activation in *C. elegans*, we examined eIF2α phosphorylation levels in wild-type and *gcn-1(n4827)* putative null (*gcn-1(−)*) animals. An increase in eIF2α phosphorylation levels that occurred under oxidizing conditions in wild-type worms was largely diminished in *gcn-1(−)* animals (Fig. 4j and k, Supplementary Fig. 5c and d). By establishing a conserved function for GCN-1 in the response of *C. elegans* to oxidative stress, our results suggest that oxidant-reactive cysteines in GCN-1 might be involved in the redox regulation of GCN-2 activity and possibly translation initiation.

**Redox-sensitive cysteines are essential for p38 activity in the antioxidant response and pathogen resistance.** Signaling through the evolutionarily-conserved MAPK p38 is activated in response to various environmental inputs, including pathogen exposure and ROS signals[10,43–45]. In *C. elegans*, the canonical p38 MAPK signaling cascade consists of NSY-1 (ASK1 MAPKKK), SEK-1 (MKK3/MKK6 MAPKK), and PMK-1 (p38 MAPK) (Fig. 5a), with PMK-1 functioning in a partially redundant manner with the related kinases PMK-2 and PMK-3[44,45]. The downstream effectors of the p38 core cassette vary among stress responses: Pathogens, such as the *Pseudomonas aeruginosa* strain PA14 trigger an innate immunity program via p38 pathway signaling by activating the transcription factor ATF-7[46], whereas the oxidative stress response is triggered by phosphorylation of the transcription factor SKN-1 (NRF2) by PMK-1[10,44].

Our site-centric proteomic analysis confirmed the presence of known oxidant-reactive cysteines in two ROS-sensing proteins in the p38 MAPK pathway: the transmembrane ER kinase/RNase IRE-1 and the MAPKKK NSY-1/ASK1[10,47]. Intriguingly, we also detected two additional oxidant-modified proteins within this pathway: SEK-1 and PMK-1 (Fig. 5a and b), and, therefore, investigated whether these newly mapped cysteines might serve as

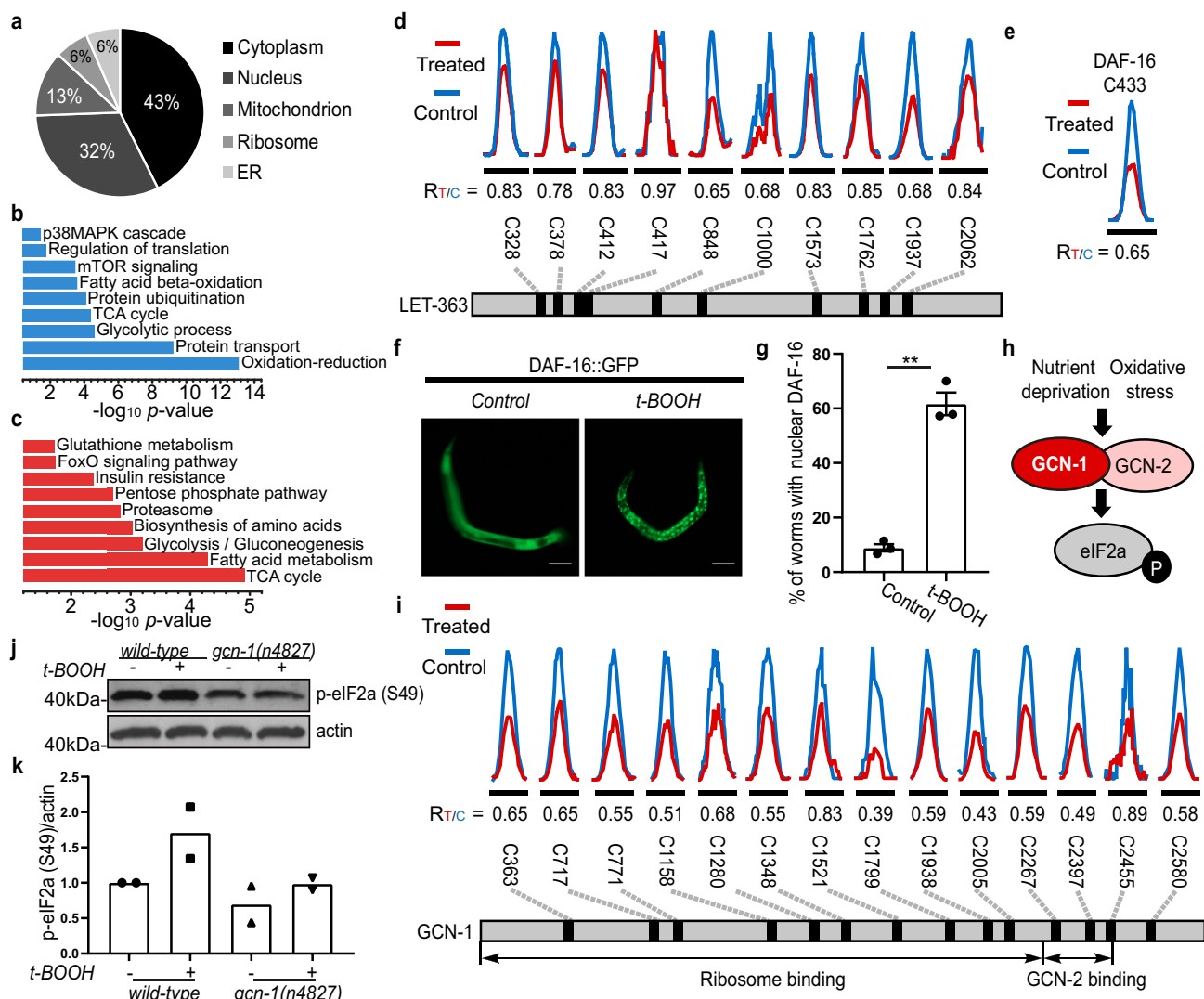

**Fig. 4 Cysteine-mediated redox regulation is involved in various biological processes and pathways. a** A pie chart showing the distribution of the 1537 proteins that exhibit at least one redox-sensitive cysteine with respect to their functions in different organelles. Subcellular localizations for *C. elegans* proteins are retrieved from the UniProt knowledge base. **b**, **c** Biological processes and pathways enriched in the *C. elegans* redoxome, indicated by GO (**b**, blue) and KEGG (**c**, red) analysis. **d**, **e** Representative XICs showing changes in IPM-labeled peptides from LET-363/mTOR (**d**) and DAF-16/FOXO (**e**). The profiles for light- and heavy-labeled peptides are shown in red and blue, respectively. The average $R_{T/C}$ values calculated from two biological replicates are displayed below individual XICs. **f** Fluorescent images and quantification **g** showing nuclear accumulation of DAF-16::GFP after 5 h of t-BOOH treatment (mean ± SEM, $n = 3$ experiments, at least 100 worms per condition). **$P = 0.0100$; Two-tailed Student's *t*-test. Scale bar = 100 μm. **h** A schematic diagram showing that various stress stimuli promote eIF2a phosphorylation by GCN-2. **i** Representative XICs showing changes in IPM-labeled peptides from GCN-1. **j** Representative western blots and quantification **k** showing that eIF2a phosphorylation in response to 30 min t-BOOH exposure is impaired in *gcn-1(−)* animals (mean, $n = 2$ experiments). Lysates were loaded onto two different gels for detection with different antibodies, and blots were processed in parallel.

redox switches in p38 MAPK-mediated stress responses and pathogen resistance.

A generally conserved cysteine within SEK-1 (C213) that we found as peroxide-reactive was of particular interest, because it is located within the kinase activation loop close to the magnesium-binding DFG motif (Fig. 5c), which is critical for protein kinase activity[10]. To investigate the importance of C213 for SEK-1 function, we used Cas9/CRISPR gene editing to introduce a conservative cysteine-to-serine mutation at C213 within the endogenous *sek-1* gene locus, and thereby prevent the SEK-1 protein from being oxidized at this residue. Importantly, phosphorylation of the p38 ortholog PMK-1 was significantly attenuated in this redox-inert *sek-1* mutant (*sek-1(syb1398)*, Fig. 5d and Supplementary Fig. 7a), indicating that the oxidation-sensitive

cysteine C213 is important for the basal level of SEK-1 kinase activity. Interestingly, while either $H_2O_2$ or the organic peroxide tert-butyl-hydroperoxide (t-BOOH) dramatically enhanced p38 phosphorylation levels in wild-type animals, such an effect was largely diminished in *sek-1* mutants (Fig. 5d and Supplementary Fig. 7a). The nuclear localization and transcriptional activity of the antioxidant response transcription factor SKN-1 depends upon its phosphorylation by phosphorylated, active PMK-1[44]. Consistent with decreased PMK-1 activity, SKN-1 activity was also lower in our *sek-1* CRISPR mutant animals under both physiological and oxidizing conditions, measured by expression of its target *gst-10* (Fig. 5f). Together, these observations indicate that the redox-sensitive cysteine C213 is critical for SEK-1 activity. It has been reported that S-sulfenylation events may increase tyrosine kinase

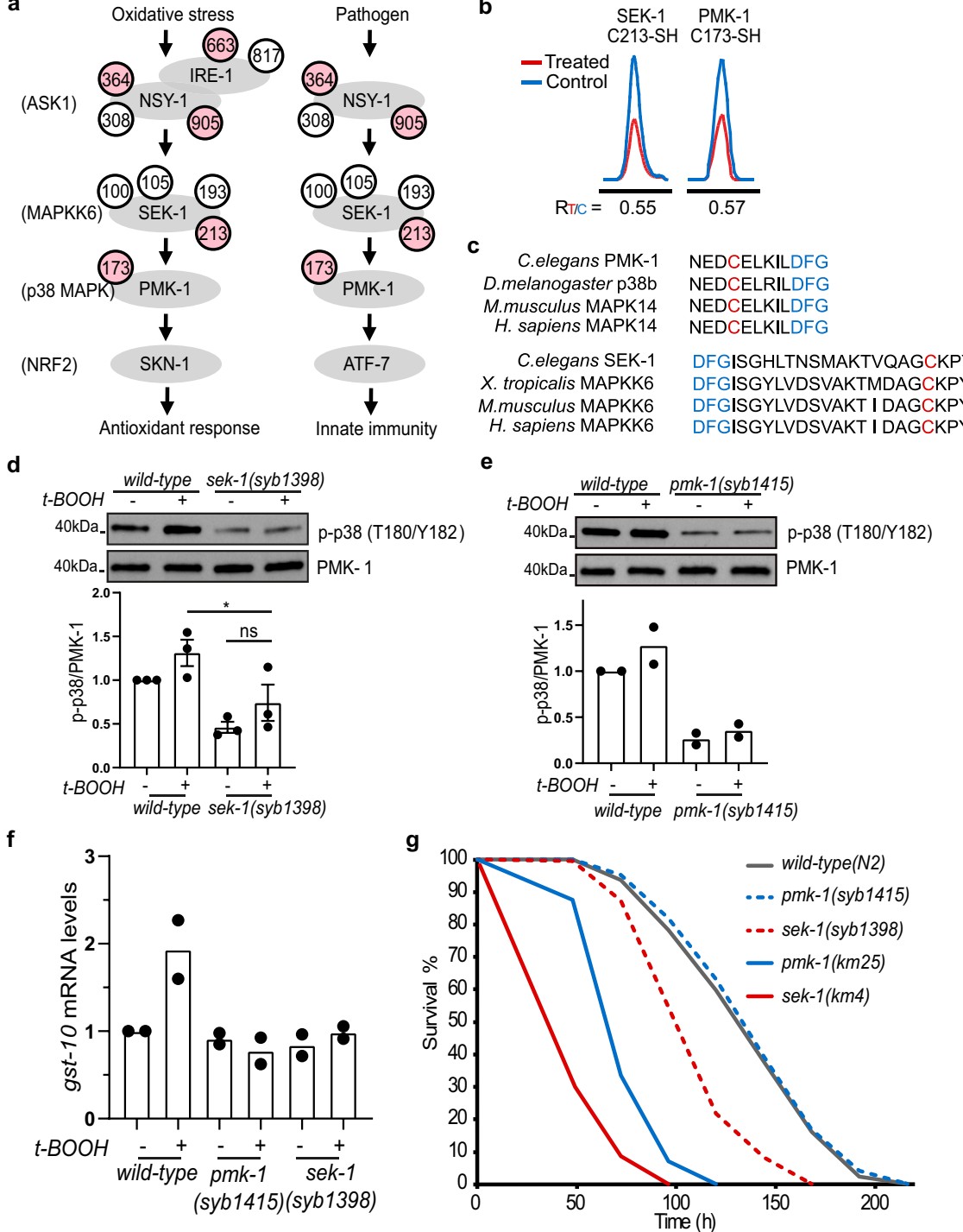

**Fig. 5 Oxidant-sensitive cysteines are essential for p38 activation. a** The p38 MAPK pathway in *C. elegans*. Cysteines in pink were identified as redox-sensitive in our analysis, while cysteines in white were not responsive to $H_2O_2$ treatment. **b** Representative XICs showing changes in IPM-labeled peptides from SEK-1 (C213) and PMK-1 (C173). The profiles for light- and heavy-labeled peptide are shown in red and blue, respectively. The average $R_{T/C}$ values calculated from two biological replicates are displayed below each XIC. **c** C213 (in red) in SEK-1 and C173 (in red) in PMK-1 are evolutionarily conserved and located close to the magnesium-binding DFG motifs (in blue). **d, e** Representative western blots in wild-type, SEK-1, and PMK-1 mutant animals carrying individual cysteine-to-serine mutations, and quantification showing that the two reactive cysteines are each important for t-BOOH-induced p38 phosphorylation (mean ± SEM, $n = 3$ experiments in (**d**); mean, $n = 2$ experiments in (**e**).) *$P = 0.0318$; ns, not significant ($P = 0.3443$); One-way ANOVA with Bonferroni post-test. Lysates were loaded onto two different gels for detection with different antibodies, and blots were processed in parallel. **f** Quantification of mRNA levels of the SKN-1 target *gst-10* by qRT-PCR in wild-type, the PMK-1 C173S mutant (*pmk-1(syb1415)*), and the SEK-1 C213S mutant (*sek-1(syb1398)*) animals with or without 30 min t-BOOH treatment (mean, $n = 2$ experiments). **g** Survival curves of wild-type animals, and *pmk-1*(−)(*km25*), *sek-1*(−)(*km4*), PMK-1 C173S (*syb1415*), and SEK-1 C213S (*syb1398*) mutants in the presence of Pseudomonas aeruginosa PA14. $n = 2$ experiments with 280–388 worms per condition. Survival summary data are provided in Supplementary Table 1.

activity[48], and our data together suggest that SEK-1 kinase activity might also be enhanced by S-sulfenylation on C213 (Supplementary Fig. 6b).

The oxidant-modified cysteine C173 in PMK-1 shows a high degree of conservation from worms to humans and is located adjacent to the magnesium-binding DFG motif within the kinase activation loop (Fig. 5c). Immunoblotting showed that a C173S mutation (pmk-1(syb1415)) largely inhibited PMK-1 phosphorylation (Fig. 5e and Supplementary Fig. 7b), without affecting PMK-1 abundance (Supplementary Fig. 7c). Importantly, PMK-1 phosphorylation levels and presumably activity were lower in pmk-1(syb1415) mutants compared to wild-type animals, under both basal and oxidative stressed conditions, implying that C173 is important for PMK-1 phosphorylation by SEK-1, and for PMK-1 activity in general (Fig. 5e, f, and Supplementary Fig. 7b). Together, the data suggest that redox-reactive cysteines in the p38 pathway are critical for basal p38 activity and p38 activation by ROS.

Having determined that cysteines in SEK-1 and PMK-1 are essential for p38 activity, we next investigated whether the same cysteines are important in the p38-mediated immune response. We evaluated survival after infection with PA14 bacteria in wild-type, sek-1, and pmk-1 mutant animals. In agreement with previous reports[45], susceptibility to PA14 infection was elevated in the presumed null mutants pmk-1(km25)(pmk-1(−)) and sek-1(km4)(sek-1(−)), with sek-1(−) mutants exhibiting more severe defects than pmk-1(−) animals (Fig. 5g and Supplementary Table 1). This observation is consistent with evidence that PMK-1 might function in a partially redundant way with its paralog PMK-2 downstream of SEK-1 in pathogen responses[49,50]. The SEK-1 C213S mutation increased pathogen sensitivity, albeit to a lesser extent than the sek-1 null mutation (Fig. 5g and Supplementary Table 1), suggesting that conservative substitution of this SEK-1 residue resulted in a partial loss of function with respect to innate immunity. By contrast, pmk-1(syb1415) animals behaved similarly to the wild-type strain with respect to PA14 survival (Fig. 5g and Supplementary Table 1), a finding that could reflect functional redundancy between PMK-1 and -2, or C173 not being essential for PMK-1 function in PA14 resistance. In summary, we conclude that the conserved amino acids PMK-1 C173 and SEK-1 C213 are both important for p38 activation by oxidative stress, with the latter also being critical for pathogen defense. Our finding that these two cysteines transduce stress signals to activate p38 implies that redox regulation occurs not only on the two known redox sensors IRE-1 and NSY-1[10,47], but literally on each component of the p38 pathway.

## Discussion

A comprehensive assessment of the protein redoxome is critical for understanding the scope of redox signaling networks in cells and organisms, and formulating models for how redox regulation might act on specific regulatory pathways and biological processes. Recent years have seen great advancements in the redox proteomics field, and studies in cells and tissues are able to detect thousands of ROS-reactive cysteines[7,18,19,24,28]. However, progress to define the redoxome in multi-cellular model organisms has not yet caught up, with the scale of redox-modified sites identified in C. elegans or D. melanogaster still being in the hundreds[13,14,20,21]. Such a large gap most likely reflects a difference in detection sensitivity and capacity, and suggests a need for comprehensive characterization of the redoxome at the organismal level.

C. elegans serves as a powerful model for studying redox signaling. It also has a relatively short lifespan (~3 weeks compared to ~3 months for flies or ~3 years for mice), a fully-sequenced genome, and contains numerous conserved cysteines in the proteome. Various tools for genetic manipulation allow the investigation of the physiological function of selective cysteines in vivo, compared to studies using cell culture models. Here we have quantified intrinsic cysteine reactivity in C. elegans at a proteome-wide scale, providing a basis for future functional screening of physiologically relevant redox events. Furthermore, we have systematically and site-specifically measured ratiometric changes in three specific redox forms (–SH, –SOH, and –SO$_2$H) in C. elegans hermaphrodites upon an acute H$_2$O$_2$ treatment. A total of 5453 –SH, 1521 –SOH, and 82 –SO$_2$H sites were identified and quantified, including numerous cysteines that were previously determined as redox-regulated (Fig. 3). Moreover, by identifying many novel redox-regulated cysteines and proteins, our results dramatically increase the scale of C. elegans redoxome data.

Many previous redox proteomics studies focused on quantifying total reversible oxidations on thiols. Within cells, thiols undergo various oxidative modifications, including –SOH, –SO$_2$H, and –SS–, which may have different effects on protein activity. For example, evidence supports that sulfinylation on C106 of the Parkinson's disease protein DJ-1 protects cells, while sulfenylation on the same cysteine does not[17]. Our study has defined the nature of many distinct oxidation events in C. elegans, by simultaneously characterizing both the reduced and oxidized forms of cysteines. We identified 1537 proteins in which at least one cysteine redox form changed dramatically (≥1.5-fold) upon oxidant treatment. PTMs such as phosphorylation and acetylation may not need to reach a very high stoichiometry to exert regulatory effects[51,52], and we expect that in many cases, the extent of fold changes in cysteine redox states is likely to be as informative as the absolute percentage modified.

By uncovering previously unappreciated oxidant-sensitive cysteines and proteins, our findings provide potential mechanistic insights into known redox-regulated biological processes. For instance, it is intriguing that we find several peroxide-sensitive sites in the mTORC1 signaling and insulin/IGF-1 signaling pathways, given that endogenous ROS levels fluctuate dramatically during early development and across the lifespan of C. elegans[6,14]. As a second example, global translation inhibition under oxidizing conditions is observed from yeast to mammalian cells[40,41,53]. Phosphorylation of the translation regulator eIF2α by the conserved protein kinase GCN-2 is involved[40,41], but it has remained unclear how ROS signals are conveyed to GCN-2. While it is possible that ROS increase GCN-1 protein levels and thereby promote eIF2α phosphorylation by GCN-2, another possibility is that ROS modulate GCN-1 activity directly. We find that the GCN-2 cofactor GCN-1 contains 11 oxidant-modified cysteines and is involved in the increase in eIF2α phosphorylation by ROS in C. elegans, suggesting that it might represent an important redox sensor that responds to environmental changes to modulate protein synthesis. We also find many redox-sensitive cysteines in translation initiation, elongation, and termination factors as well as ribosomal proteins (Supplementary Data 6), suggesting that decreased protein synthesis under oxidative stress might be controlled at multiple steps of translation.

Gene expression patterns are coordinated by transcription, epigenetic, and mRNA splicing factors. Under oxidizing conditions, the activation of sequence-specific transcription factors such as SKN-1/NRF and DAF-16/FOXO alters expression levels of target genes[10,38,54]. Recently it has been reported that ROS modify SET-1/MLL1 histone methyltransferases and thereby decrease global H3K4me3 levels[6]. We find additional chromatin-modifying enzymes in the C. elegans redoxome, suggesting that potential changes in other epigenetic marks might also occur. Splicing factors are subject to post-translational modifications such as phosphorylation and acetylation, which affect their localization

and activity[55]. Although pre-mRNA splicing has not been implicated in redox signaling before, in this study we identify several regulatory factors of splicing as apparent ROS targets (Supplementary Data 6). In summary, our analyses indicate that the redox regulation of gene expression and mRNA levels is likely to be achieved at both transcriptional and post-transcriptional levels.

We determined that cysteines in SEK-1 and PMK-1 that we identified as redox-modified are each important for p38 phosphorylation and activation, indicating redox regulation of p38 signaling at each step in the pathway (Fig. 5 and Supplementary Fig. 7). Given that oxidative modifications can alter protein conformation or ligand binding affinity[1], an intriguing possibility is that cysteine oxidation in SEK-1 and PMK-1 might promote the interaction between SEK-1 and PMK-1, and thus increase PMK-1 phosphorylation levels. It is also likely that oxidation enhances SEK-1 kinase activity through other mechanisms, such as protein stability or subcellular localization. Our results suggest that oxidation and activation of the entire p38 MAPK signaling cascade might occur near the ER membrane, where NSY-1 is localized during activation[10].

During mammalian innate immunity responses, increased ROS production activates various adaptive mechanisms within phagocytes and thus contributes to pathogen elimination[56]. In C. elegans, pathogen exposure leads to ROS production by the NADPH oxidase BLI-3, and thereby activates p38 MAPK signaling[57], an ancient and conserved immune response mechanism[45]. We find that oxidation on a conserved cysteine in the kinase activation loop of SEK-1 is essential for pathogen defense in C. elegans (Fig. 5), suggesting a role for redox signaling in pathogen resistance that involves p38 signaling. Such observation also suggests that post-translational oxidation on cysteines plays a broader role in stresses besides oxidative stress defense. Given that the functionally important redox-regulated cysteines we identified in SEK-1 and PMK-1 are conserved (Fig. 5), in the future it will be important to explore in more complex organisms how redox regulation of individual signaling components influences p38 pathway activation in response to different stimuli and in different cell types.

In summary, our quantitative, site-centric map of cysteine intrinsic reactivity and dataset of peroxide-sensitive cysteine redox forms greatly expand the scope and biological role for cysteine oxidation in C. elegans, and provides a molecular basis to decipher the complicated redox signaling networks in the model organism. It will stimulate future investigations to uncover the biological significance of individual cysteines in C. elegans and other organisms where they are conserved. As cellular redox states and cysteine modification events are altered during aging and in several diseases[7,8,14], understanding of cysteine redox profiles and redox states of disease-driving proteins under physiological and pathological conditions may also provide insights about potential therapeutic strategies.

## Methods

**Reagents.** BTD and DiaAlk were homemade; Catalase (C9322), Iodoacetamide (V900335), Tris[(1-benzyl-1H-1,2,3-triazol-4-yl)methyl]amine (TBTA) (678937), and sodium ascobate (A7631) were purchased from Sigma-Aldrich; dithiothreitol (A100281) was purchased from BBI Life Sciences; IPM (EVU111), Light Azido-UV-Biotin (EVU102), and Heavy Azido-UV-Biotin (EVU151) were purchased from KeraFast; CuSO₄ (C493-500) was purchased from Thermo Fisher Scientific; Sequencing grade trypsin (V5113) was purchased from Promega. Anti-actin antibody (A5441) (1:500) and anti-tubulin antibody (T9026) (1:500) were purchased from Sigma-Aldrich; Anti-p-eIF2α (3597) (1:1000) and anti-p-p38 (9211) (1:500) antibodies were purchased from Cell Signaling Technology; Anti-PMK-1 antibody (1:500) was a kind gift from Dr. R. Pukkila-Worley at University of Massachusetts Medical School, Worcester, MA[58]; Anti-PRDX-2 antibody (1:2000) was a kind gift from Dr. E. Veal at Newcastle University Biosciences Institute, Newcastle upon Tyne, UK[30].

**Oligonucleotides.** All primers used to amplify specific regions from the genome, or to measure mRNA levels by qRT-PCR are listed in Supplementary Table 2. sgRNAs and repair templates to generate the alleles pmk-1(syb1415) and sek-1 (syb1398) are also listed in Supplementary Table 2.

**Bacterial strains.** Escherichia coli (E. coli) strain OP50-1 was from the Caenorhabditis Genetics Center (#OP50-1; RRID: WB-STRAIN:OP50-1); E. coli strain JI377 was a gift from Dr. J. A. Imlay at University of Illinois at Urbana-Champaign, Urbana, IL[59]; Pseudomonas aeruginosa strain PA14 was a gift from Dr. D. Kim at Harvard Medical School, Boston, MA.

**Maintenance of C. elegans.** The following C. elegans strains were acquired from the Caenorhabditis Genetics Center: wild-type (N2) and TJ356: zIs356(daf-16p:: daf-16a/b::GFP;rol-6(su1006)). LD1758 was obtained by backcrossing pmk-1 (km25) 5× to N2, whereas LD1757 was obtained by backcrossing sek-1(km4) 4X to N2. Alleles pmk-1(syb1415) and sek-1(syb1398) were generated by SunyBiotech (China), and verified by us with sequencing. LD1906 was obtained by backcrossing pmk-1(syb1415) 4X to N2, and LD1886 was obtained by backcrossing sek-1 (syb1398) 4X to N2. MT22914 was acquired from Dr. H.R. Horvitz at MIT, Cambridge, MA[60]. Worms were maintained on NGM plates seeded with OP50-1 at 20 °C.

**Preparation of protein lysates for proteomics.** Wild-type C. elegans at the L4 stage were cultured in the presence of 0.05 g/L FUDR to prevent reproduction. After 3 days of growth, worms were harvested, washed with pre-chilled PBS buffer (KH₂PO₄ 30 g/L, K₂HPO₄ 60 g/L, NaCl 50 g/L) for three times, and then incubated with or without 5 mM H₂O₂ for 5 min. After being washed with PBS containing 200 unit/mL catalase for three times, C. elegans pellets were lysed in four volumes of pre-chilled NETN buffer (50 mM HEPES (pH = 7.6), 150 mM NaCl, and 1% IGEPAL) supplemented with 1× protease and phosphatase inhibitors (Thermo Fisher Scientific, A32961) containing 200 unit/mL catalase (Sigma-Aldrich) using Retsch homogenizer (Retsch GmbH, MM400).

**Probe labeling.** For intrinsic cysteine reactivity, protein lysates were separated into two identical aliquots and incubated with either 10 μM or 100 μM IPM at room temperature (RT, ~25 °C) for 1 h with rotation and light protection. For SH labeling, protein lysates were incubated with 100 μM IPM at RT for 1 h with rotation and light protection[23]. For –SOH, protein lysates were incubated with 5 mM BTD at 37 °C for 1 h with rotation[29]. For –SO₂H, protein lysates were incubated with 2.5 mM DPS (Sigma-Aldrich, D5767) at RT for 1 h with rotation and quenched by protein precipitation with a pre-chilled methanol/chloroform system (~4:4:1 (vol/vol/vol) lysate/methanol/chloroform). The precipitated proteins were washed twice with 500 μL methanol and resuspended with 800 μl of PBS containing 0.5% SDS, and then labeled with 5 mM DiaAlk probe at 37 °C for 2 h with rotation and light protection.

**Preparation of the probe-labeled protein samples.** The probe-labeled protein samples were incubated with 10 mM DTT at RT for 1 h, followed by incubation with 40 mM iodoacetamide at RT for 1 h with light protection. Proteins were then precipitated with a methanol–chloroform system (aqueous phase/methanol/ chloroform, 4:4:1 (v/v/v)) as previously described[29]. The precipitated proteins were resuspended with 50 mM ammonium bicarbonate containing 0.2 M urea. Resuspended proteins were digested with sequencing grade trypsin (Promega) at a 1:50 (enzyme/substrate) ratio overnight at 37 °C. The tryptic digests were desalted with HLB extraction cartridges (Waters) and dried under vacuum. In a solution containing 30% acetonitrile (MeCN) at pH ~6, CuAAC reaction was performed by subsequently adding 1 mM either light or heavy Azido-UV-biotin (1 μL of a 40 mM stock), 10 mM sodium ascorbate (4 μL of a 100 mM stock), 1 mM TBTA (1 μL of a 50 mM stock), and 10 mM CuSO₄ (4 μL of a 100 mM stock). Samples were allowed to react at RT for 2 h with rotation and light protection. The light and heavy isotopic tagged samples were then mixed immediately following click chemistry. The samples were cleaned by strong cation exchange (SCX, Nest group, cat. No SMM HIL-SCX) spin columns and then subject to the enrichment with streptavidin beads for 2 h at RT. Streptavidin beads were washed with 50 mM NaAc (pH = 4.5), 50 mM NaAc containing 2 M NaCl (pH = 4.5), and deionized water twice each with vortexing and/or rotation to remove non-specific binding substances, then resuspended in 25 mM ammonium bicarbonate, transferred to glass tubes (VWR), and irradiated with 365 nm UV light (Entela, Upland, CA) for 2 h at RT with magnetic stirring. The supernatant was collected, concentrated under vacuum, and desalted with HLB cartridges. The resulting peptides were evaporated to dryness and stored at −20 °C until LC-MS/MS analysis.

**Liquid chromatography-tandem mass spectrometry (LC-MS/MS) analysis.** LC-MS/MS analyses were performed on a Q Exactive plus instrument (Thermo Fisher Scientific) or a Q Exactive HF instrument (Thermo Fisher Scientific). Peptide samples were reconstituted in 0.1% formic acid and pressure-loaded onto a 2-cm microcapillary precolumn packed with C18 (3-μm, 120 Å, SunChrom, USA) operated with an Easy-nLC1000 system (Thermo Fisher Scientific). The precolumn

was connected to a 12-cm 150-µm-inner diameter microcapillary analytical column packed with C18 (1.9-µm, 120 Å, Dr. Maisch GebH, Germany) and equipped with a homemade electrospray emitter tip. The spray voltage was set to 2.0 kV and the heated capillary temperature to 320 °C. LC gradient consisted of 0 min, 7% B; 14 min, 10% B; 51 min, 20% B; 68 min, 30% B; 69-75 min, 95% B (A = water, 0.1% formic acid; B = MeCN, 0.1% formic acid) at a flow rate of 600 nL/min.

For Q Exactive Plus, MS1 spectra were recorded with a resolution of 70,000, an AGC target of 3e6, a max injection time of 20 ms, and a mass range from $m/z$ 300 to 1400. HCD MS/MS spectra were acquired with a resolution of 17,500, an AGC target of 1e6, a max injection time of 60 ms, a 1.6 $m/z$ isolation window and normalized collision energy of 30. Peptide $m/z$ that triggered MS/MS scans were dynamically excluded from further MS/MS scans for 18 s.

For Q Exactive HF, MS1 spectra were recorded with a resolution of 120,000, an AGC target of 3e6, a max injection time of 80 ms, and a mass range from $m/z$ 300 to 1400. HCD MS/MS spectra were acquired with a resolution of 15,000, an AGC target of 5e4, a max injection time of 20 ms, a 1.6 $m/z$ isolation window and normalized collision energy of 27. Peptide m/z that triggered MS/MS scans were dynamically excluded from further MS/MS scans for 12 s.

**Peptide identification and quantification**. Raw data files were searched against *C. elegans* Uniprot canonical database (https://www.uniprot.org/proteomes/UP000001940). Database searches were performed with pFind studio (Version 3.0.11, http://pfind.ict.ac.cn/software/pFind3/index.html)[61]. Precursor ion mass and fragmentation tolerance were set as 10 ppm and 20 ppm, respectively. The maximum number of modifications allowed per peptide was three, as was the maximum number of missed cleavages allowed. For all analyses, mass shifts of +15.9949 Da (methionine oxidation) and +57.0214 Da (iodoacetamide alkylation) were searched as variable modifications. For site-specific mapping of probe-modified –SH, –SOH, and –SO₂H sites, mass shifts of +252.1222 (light IPM-triazohexanoic acid), +418.1311 (light BTD-triazohexanoic acid), and +387.1754 (light DiaAlk-triazohexanoic acid) were searched as variable modifications, respectively. A differential modification of 6.0201 Da on probe-derived modification was used for stable-isotopic quantification. The FDRs were estimated by the program from the number and quality of spectral matches to the decoy database. The FDRs at spectrum, peptide, and protein level were <1%. Quantification of heavy to light ratios ($R_{H/L}$) was performed using pQuant as previously described[62], which directly uses the RAW files as the input. pQuant calculated $R_{H/L}$ values based on each identified MS scan with a 15 ppm-level $m/z$ tolerance window and assigned an interference score (Int. Score, also known as confidence score) to each value from zero to one. In principle, the lower the calculated Int. Score was, the less co-elution interference signal was observed in the extracted ion chromatograms. In this regard, the median values of probe-modified peptide ratios with σ less than or equal 0.5 were considered to calculate site-level ratios. Quantification results were obtained from two or three biological replicates with single 75-min LC-MS/MS run for each and were reported in Excel files (Supplementary Data 1-4).

**Bioinformatics**. The *C. elegans* redoxome was subjected to enrichment analyses using the web portal bioinformatics tool DAVID with terms including Gene Ontology (GO) cellular compartments, GO biological processes, and Kyoto Encyclopedia of Genes and Genomes (KEGG) pathways[63]. Scatter plots were made by Origin (version 8). Pie charts and bar charts were made by Excel 2016. The violin plot was made by BoxPlotR[64].

**Stress treatments**. L4 worms were suspended in M9 solution containing either 1 mM t-BOOH (Sigma) or 1 mM hydrogen peroxide (H₂O₂) (Sigma) and rotated in the presence of food for 30 min at 20 °C, unless otherwise specified. A lower dose of oxidants was used in these experiments than in the proteomic analyses in order to prevent animals from being over-stressed during the longer time course that was required. For t-BOOH treatments, OP50-1 was used as food source. For H₂O₂ treatments, the *KatG KatE AhpCF* triple null mutant bacterial strain JI377, which cannot scavenge hydrogen peroxide from the environment, was used as the food source[65].

**Fluorescence microscopy**. Images were collected using a ×20 objective on an OLYMPUS IX51 microscope with Olympus Cell Sens Standard.

**Immunoblot analysis**. To detect oxidation of PRDX-2 (Supplementary Fig. 3), synchronized *C. elegans* were harvested at the same stage and lysed using the same lysis buffer as for proteomic analyses. Proteins were separated on non-reducing SDS-PAGE gels. For all other immunoblots, synchronized *C. elegans* at L4 stage were washed with M9 for at least three times. After the last wash, 6,000 worms were sonicated in 100 µl of lysis buffer (50 mM Tris-HCl (pH 7.5), 150 mM NaCl, 1% (v/v) NP-40, 2 mM EDTA) containing protease inhibitor and phosphatase inhibitor cocktails (Roche), and insoluble material was removed by centrifugation for 6 min at 13,000 × g at 4 °C. For equal loading, protein concentrations of each sample were determined with the Pierce™ BCA Protein Assay Kit and a standard curve with Bovine Serum Albumin (Thermo Fisher Scientific). Protein samples were boiled at 95 °C for 5 min. Immunoblotting was performed with 4-15% gradient Mini-PROTEIN TGX precast polyacrylamide gels and nitrocellulose membranes (Bio-Rad). Blots were blocked with 5% milk for 1 h, and antibodies were incubated with 5% milk or bovine serum albumin in TBS with 0.1% Tween 20. Chemiluminescent detection of HRP signals was performed via a VersaDoc imaging station (Bio-Rad) or an Epson Perfection scanner. All immunoblots are representative of at least two experiments. The relative abundances of bands were quantified by ImageJ (Version: Java 1.6.0_24 (64-bit)). Uncropped blots are provided in the Source data file.

**qRT-PCR assays**. For each sample, ~3000 L4 worms were used, and total RNA was extracted with TRI REAGENT (Sigma) and the Direct-zol RNA Miniprep Plus Kit (ZYMO Research). First-strand cDNA was synthesized using the Applied Biosystems™ High-Capacity cDNA Reverse Transcription Kit (Thermo Fisher Scientific). SYBR green (Thermo Fisher Scientific) was used to perform qRT-PCR (ABI 7900). mRNA levels were normalized by using *act-1* as a reference gene. Two independent biological replicates were examined for each sample. Gene expression fold change was calculated using the ΔΔCt method.

**Pathogenesis assays**. Single colonies of PA14 were cultured overnight at 37 °C in LB. 5 µl of the overnight culture were seeded onto the center of a 35 mm NGM agar plate containing 0.02 g/L FUdR. Plates were incubated overnight at 37 °C before being incubated overnight at room temperature. About 40 worms were transferred to each plate, with 5 plates scored per condition. The assays were conducted at 20 °C. Animals that did not respond to gentle prodding from a platinum wire were scored as dead.

**Statistical analysis**. Statistical tests and *n* values used are indicated in the corresponding figure legends. All data were analyzed using GraphPad Prism version 8.4.2 software.

**Reporting summary**. Further information on research design is available in the Nature Research Reporting Summary linked to this article.

## Data availability

The data that support this work is available from the corresponding authors upon reasonable request. The mass spectrometry proteomics data have been deposited to the ProteomeXchange Consortium via the PRIDE partner repository with project accession PXD018575. Source data are provided with this paper.

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

## Acknowledgements

We thank Dr. Read Pukkila-Worley, Dr. Elizabeth Veal, Dr. Javier Apfeld, and Dr. H. Robert Horvitz for reagents; Dr. Meng-Qiu Dong from National Institute of Biological Sciences, Beijing, for her advice and help on preparing *C. elegans* samples for proteomics; Longqin Sun and Tuo Zhang from Beijing Qinglian Biotech Co., Ltd for their help and technical supports. The work was supported by grants from the National Key R&D Program of China (2016YFA0501303) to J.Y., the Natural Science Foundation of China (21922702, 81973279, and 31770885) to J.Y. and (31800036) to L.F., the State Key Laboratory of Proteomics (SKLP-K201703 and SKLP-K201804) to J.Y., and the NIH to T.K.B. (R35 GM122610), along with a DRC grant from the NIDDK (P30 DK036836). J.M. was supported by the Iacocca Family Foundation. Some strains were provided by the Caenorhabditis Genetics Center, which is funded by the NIH (P40 OD010440). The authors T. Keith Blackwell and Jing Yang jointly supervised this work.

## Author contributions

Conceptualization, J.Y. and T.K.B.; methodology, J.M., L.F., and J.Y.; investigation, J.M., L.F., K.L., C.T., and Z.W.; resources, Y.J., R.B.F., and K.S.C.; writing—original draft, J.M. and J.Y.; writing—review and editing, J.M., J.Y., and T.K.B.; funding acquisition, J.Y., L.F., J.M., and T.K.B.; supervision, J.Y. and T.K.B.

## Competing interests

The authors declare no competing interests.
