## [Peer Review File · Nature Communications]

Reviewer #1 (Remarks to the Author):

It has been proposed that the sensitivity of cysteine to irreversible oxidation has imposed an evolutionary pressure, limiting their abundance on the surface of proteins to functional cysteines. However, there are also well-characterized examples where reversible oxidation of cysteines is used to regulate protein activity. Hence, a quantitative determination of which cysteines in the proteome are more sensitive to oxidation is likely to identify new functional cysteines that are candidates for redox regulation. Here Meng et al have used a sophisticated MS approach, utilizing clickable thiol-reactive tools, used previously, to identify reactive and H₂O₂-oxidizable cysteines in the *C. elegans* proteome. Their study builds on previous studies using similar (Martell et al 2016) and different (Kumsta et al 2011&Knoepfler et al 2012) global approaches to identify redox-sensitive cysteines in *C. elegans*. However, their work extends these studies significantly by revealing many new candidate peroxide-regulated cysteine thiols, including in less-abundant signaling proteins. It also differentiates between different cysteine oxidation states: sulfenylation and sulfinylation.

As with any global study, the challenge is to demonstrate physiological relevance of the oxidation of identified reactive cysteines. In this respect, the authors present data to suggest that the observed oxidation-sensitivity of cysteines in GCN1 might reflect a role for GCN1 oxidation in inhibiting translation by promoting the peroxide-induced phosphorylation of eif2 α . They also identify oxidation-sensitive cysteines in a MAPKK(SEK-1) and a conserved MAPK(PMK-1) that are important for activation of PMK-1 and function of SEK-1.

This work has the potential to be an important resource for other researchers in redox/stress signaling and translation fields. However, some aspects need further clarification/explanation, or more cautious interpretation, before I could be fully supportive of publication.

[1] The data presented suggest that Meng et al's approach to detecting reactive cysteines is robust and reproducible. However, if the hyper-reactivity of these cysteines is taken to indicate their susceptibility to oxidation, then, as they indicate, it is surprising that cysteines that are known to be highly reactive with peroxides, such as catalytic cysteines in thioredoxin and peroxiredoxin proteins, were only detected as moderately or relatively unreactive (Extended dataset 1). Moreover, these proteins do not seem to present in the sulfinylated proteins (Extended dataset 4). This is particularly surprising for the peroxide-reactive cysteine in the PRDX-2, that has previously been shown to be oxidized in these conditions (Olahova et al 2008, Thamsen et al 2011). Do the authors think this means that there are a lot of protein cysteine-thiols that are more reactive than the catalytic cysteines in these peroxidases, or is there some other explanation for why their study does not detect that these cysteines are hyper-reactive and sensitive to oxidation?

Similarly, as the authors note, the most reactive cysteine in UBA-1 was not its catalytic cysteine, and catalytic cysteines in GAPDH(Gpd) which have been shown to be sensitive to oxidation by H₂O₂ (Peralta et al 2015) were lower in reactivity than many other cysteines. Again, this raises questions as to how confident we can be that their approach identifies quantitative differences in biological

reactivity? For example, do values of 3 or 6, warrant a general definition as moderately or hyper-reactive? -Each approach has different strengths and limitations. It would be helpful if the authors could provide further justification for these definitions or else, explain how their findings should be viewed in the context of other studies?

[2] The authors have detected cysteines in various components of the MAPK pathway that are sensitive to oxidation. To test whether this oxidation might be important for the function of 2 of these kinases (MAPKK, SEK-1 and MAPK, PMK-1), they have generated *C. elegans* expressing Cys to Ser mutant versions of each. Analysis of these strains suggests the oxidation-sensitive cysteines are important for the phosphorylation of PMK-1. Indeed, assuming that SEK-1 mutant protein levels are unaffected (which should be shown), then their data provide strong evidence that C213 is important for the kinase activity and biological function of SEK-1 in protecting against infection (Fig. 5d, f and h). However, the presented data do not support the conclusion that oxidation of either cysteine is responsible for the increased PMK-1P in response to peroxide: The largest effect of each cysteine substitution (Fig. 5d-g) seems to be on the levels of PMK-1P under physiological conditions, with H₂O₂ and tBOOH both inducing increased PMK-1P in animals bearing either the *sek-1* or *pmk-1* mutant alleles. This suggests that the reduced form of these cysteines are important for PMK-1 phosphorylation. Moreover, animals expressing the PMK-1 C173S mutant seem to have no reduction in their immunity to PA14. This suggests that, although required for PMK-1 phosphorylation, this cysteine is not important for PMK-1's innate immune function.

[3] p. 3 The authors' proteomic studies were carried out with 5mM H₂O₂. Prolonged exposure to 5-10mM H₂O₂ is lethal to *C. elegans* (Kumsta et al 2011, Olahova et al 2008) so it seems misleading to describe this as 'low dose' H₂O₂. Other experiments were carried out with lower levels (1mM) so some explanation of the choice of concentrations for different experiments would be helpful?

[4] Although the authors used H₂O₂ in their MS studies, in some of their subsequent expts, eg. (Fig.4) DAF-16 localisation and the effect of GCN-1 on p-eIF2 α , they use tBOOH rather than H₂O₂? Some explanation for the altered choice of peroxide should be included or these experiments repeated with H₂O₂.

[5] Line 141 It is well established that the structural environment will influence the sensitivity of particular cysteines to oxidation. Therefore, it is not clear to me why the authors consider it intriguing that different cysteines in the same protein showed variable reactivity?

[6] line148 -The authors follow up describing how the active site cysteine is less reactive in one of the 4 GAPDH parologs with 'we also noticed that the inherent reactivity of certain cysteines remained relatively constant through evolution'. Some explanation of this apparent contradiction is needed: Perhaps the point they are making is that their analysis does not reveal general rules, but examples of both conserved and non-conserved reactivity?

Reviewer #2 (Remarks to the Author):

This manuscript describes the use of a suite of redox probes that assess cysteine SH, SOH and SO₂H in *C. elegans* treated with hydrogen peroxide to induce oxidative stress. The authors first identify hyperreactive cysteines in *C. elegans* lysates, and then proceed to quantifying changes in SH, SOH and SO₂H upon peroxide treatment. In general, the data show that SO₂H forms show the most dramatic increases upon peroxide treatment. The authors present a compendium of redox sensitive cysteines in *C. elegans* and identify pathways that are likely to be redox regulated. In particular, they focus on the MAPK signaling pathway and identify two cysteines on PMK1 and and SEK1 that regulated downstream p38 activation.

This study provides a useful database of redox sensitive cysteines in *C. elegans* as a resource to the redox biology community. I recommend publication upon addressing the comments below:

1. For the mass spectrometry data provided, it is unclear how (or if) the data were filtered to obtain the final list. For example, the data should be filtered to only include those hits that provide ratios in 2 of 2, or 2 of 3 replicates, with hits that show high standard deviation removed. This would provide a more high confidence dataset for the community than the current data, which provides hits based on an effective n=1 ratio value.
2. Figure 4f and g, what concentrations and times were used for the tBuOOH treatment? Do these conditions mimic that used for the initial proteomics studies i.e. 5mM H₂O₂ for 5 mins?
3. Figure 3g, it is unclear if there is a clear bias for oxidation of odd numbered cysteines given the data provided. What is the percentage of even vs. odd cysteines for the entire proteome? Is this ratio skewed when looking at the pool of oxidized cysteines? In general, I felt the data does not strongly support this conclusion, and recommend removal of this section if the proteome-wide versus oxidized-cysteine ratios for odd versus even do not change significantly.
4. Figure 5h, cysteine mutation in pmk-1 has no effect on survival, whereas the pmk-1 null mutants show a distinct decrease in survival. The authors claim that this is due to functional redundancy with pmk-2. However, the presented data also supports the conclusion that this cysteine is not important for pmk-1 function. A stronger case needs to be made to convince the readers that the lack of change is due to pmk-2 redundancy instead of the cysteine just not being functionally relevant.

Reviewer #3 (Remarks to the Author):

In this manuscript, Meng et al. characterize the redoxome of *C. elegans* by applying a redox-proteomic approach that enables the measurement of changes in three major cysteine redox forms after hydrogen peroxide treatment. The authors are able to assess the redox state of ~5000 cysteines, a several fold improvement compared to previous studies. They identify novel redox-

sensitive cysteine residues, and generate redox-inactive worm mutants for some of these new candidates.

Overall, the manuscript is clearly written and logically structured, and the figures are well presented. The study represents a valuable advance to the redox field, as well as appealing to broader audiences interested in proteomics, post-translational modifications, oxidative stress responses and cell signaling pathways. The full redoxome dataset generated will be an important resource for future studies and discoveries.

The following minor comments should be addressed before publication:

Figure 4

1) The authors find that GCN1 is widely oxidized in response to H₂O₂ treatment. In panels 4j/k, they show that oxidative stress by t-BOOH does not result in enhanced phosphorylation of eIF2 α in GCN-1-null worms as seen in WT controls. The authors suggest that oxidant-reactive cysteines in GCN-1 might be involved in the redox regulation of GCN-2 activity. In this context, it would be helpful to know whether the levels of GCN-1 are altered in response to t-BOOH in WT worms?

Figure 5

2) In panels 5d-g, it may be useful to have 'C213S' and 'C173S' along with the sek-1(syb1398) and pmk-1(syb1415) labels to clearly indicate that these are redox knock-in mutants, as this information is in the main text but not the figure legend text. The same point is valid for the survival curve in panel 5h, to distinguish more explicitly between the redox mutants and the null mutants.

3) On page 10 of the manuscript: "Immunoblotting showed that a C173S mutation (pmk-1(syb1415)) did not affect PMK-1 protein abundance (Fig. 5e and 5g)." This refers to the bands labelled as p38 on the blots, which may lead to slight confusion for the reader. Also is there an accompanying actin or other loading control?

4) In panels 5d/f, the authors show that the SEK-1 C213S mutant results in diminished phosphorylation of p38 compared to its WT counterpart in response to t-BOOH and H₂O₂. However, the phospho-p38/p38 levels are also lower in the redox-inert mutant even under untreated conditions. Does this mean that C213 is not only necessary for modulation of SEK-1 activity in response to oxidative stress but also for its general activity? The SEK-1 C213S mutant still seems to be responding to oxidative stress at the level of p38 phosphorylation i.e. the phospho-p38/p38 response ratio upon oxidative treatment relative to the respective untreated controls appears to be similar - is this the case?

5) As shown by the authors for PMK-1, is SEK-1 protein stability/abundance affected by the C213S mutation?

We thank the reviewers for the time they devoted to evaluating our manuscript. Their suggestions were very helpful, and we have done our best to address their questions and concerns in the revised paper.

Reviewer #1 (Remarks to the Author):

It has been proposed that the sensitivity of cysteine to irreversible oxidation has imposed an evolutionary pressure, limiting their abundance on the surface of proteins to functional cysteines. However, there are also well-characterized examples where reversible oxidation of cysteines is used to regulate protein activity. Hence, a quantitative determination of which cysteines in the proteome are more sensitive to oxidation is likely to identify new functional cysteines that are candidates for redox regulation. Here Meng et al have used a sophisticated MS approach, utilizing clickable thiol-reactive tools, used previously, to identify reactive and H₂O₂-oxidizable cysteines in the *C. elegans* proteome. Their study builds on previous studies using similar (Martell et al 2016) and different (Kumsta et al 2011&Knoepfler et al 2012) global approaches to identify redox-sensitive cysteines in *C. elegans*. However, their work extends these studies significantly by revealing many new candidate peroxide-regulated cysteine thiols, including in less-abundant signaling proteins. It also differentiates between different cysteine oxidation states: sulfenylation and sulfinylation.

As with any global study, the challenge is to demonstrate physiological relevance of the oxidation of identified reactive cysteines. In this respect, the authors present data to suggest that the observed oxidation-sensitivity of cysteines in GCN1 might reflect a role for GCN1 oxidation in inhibiting translation by promoting the peroxide-induced phosphorylation of eif2a. They also identify oxidation-sensitive cysteines in a MAPKK(SEK-1) and a conserved MAPK(PMK-1) that are important for activation of PMK-1 and function of SEK-1.

This work has the potential to be an important resource for other researchers in redox/stress signaling and translation fields. However, some aspects need further clarification/explanation, or more cautious interpretation, before I could be fully supportive of publication.

Response: We thank the reviewer for the enthusiastic comments regarding the main points of our work.

[1] The data presented suggest that Meng et al's approach to detecting reactive cysteines is robust and reproducible. However, if the hyper-reactivity of these cysteines is taken to indicate their susceptibility to oxidation, then, as they indicate, it is surprising that cysteines that are known to be highly reactive with peroxides, such as catalytic cysteines in thioredoxin and peroxiredoxin proteins, were only detected as moderately or relatively

unreactive (Extended dataset 1). Moreover, these proteins do not seem to present in the sulfenylated proteins (Extended dataset 4). This is particularly surprising for the peroxide-reactive cysteine in the PRDX-2, that has previously been shown to be oxidized in these conditions (Olahova et al 2008, Thamsen et al 2011). Do the authors think this means that there are a lot of protein cysteine-thiols that are more reactive than the catalytic cysteines in these peroxidases, or is there some other explanation for why their study does not detect that these cysteines are hyper-reactive and sensitive to oxidation?

Similarly, as the authors note, the most reactive cysteine in UBA-1 was not its catalytic cysteine, and catalytic cysteines in GAPDH(Gpd) which have been shown to be sensitive to oxidation by H₂O₂ (Peralta et al 2015) were lower in reactivity than many other cysteines. Again, this raises questions as to how confident we can be that their approach identifies quantitative differences in biological reactivity? For example, do values of 3 or 6, warrant a general definition as moderately or hyper-reactive? -Each approach has different strengths and limitations. It would be helpful if the authors could provide further justification for these definitions or else, explain how their findings should be viewed in the context of other studies?

Response: We thank the reviewer for raising such important concerns. As is outlined in the manuscript text and below, we see our reactivity profiling as providing a very valuable resource that is complementary to our H₂O₂-sensitivity analyses. At the same time, we have rewritten the first section of Results in the manuscript from the ground up (Page 5-6) in order to provide a broader view of its potential limitations.

Firstly, we would like to clarify that the intrinsic reactivity of a cysteine does not always correlate with its susceptibility to oxidation. The intrinsic reactivity of each measured cysteine is determined by its reaction kinetics with the electrophilic probe IPM, which may not correspond exactly with its interactions with ROS (e.g., H₂O₂). This explains why cysteines that were 'intrinsically' reactive were not always reactive with peroxides, *vice versa*. In concert with this observation, similar results have been reported in human cells, where only ~2.3% of the H₂O₂-sensitive cysteines exhibited high or moderate reactivity (Fu et al., *Mol Cell Proteomics*, 2017; Weerapana et al., *Nature*, 2010). Thus, it is not unexpected that peroxide-sensitive cysteines in GAPDH/GPDs or peroxiredoxin proteins did not show very high measured reactivity.

As the reviewer pointed out about PRDX-2, we did not recover all cysteines that are known to be oxidized/sulfenylated in the sulfenylation dataset. One possible reason is that the peptides of PRDXs generated by trypsin digestion containing such cysteines are too long (~25 AAs) and too hydrophobic, limiting their LC-MS/MS detection. In fact, peroxidatic sites of PRDXs have never been detected in any of our datasets generated from proteomic analyses in different species (e.g., human, flies, plants). Besides, the oxidation state of cysteines is spatially and temporally controlled, and could be affected by a protein's subcellular localization and ROS levels. This might also explain the lack of

some known redox-sensitive sites responded to the H₂O₂ treatment. As a result, cysteines missing in our redoxome dataset are not necessarily less peroxide-sensitive than those profiled.

As the reviewer also noted, several ‘intrinsically hyper-reactive’ cysteines including the cysteine in UBA-1 have not been functionally annotated, while a number of known active or redox-sensitive cysteines only exhibited moderate reactivity. Although some hyper-reactive cysteines with a low R_{100:10} value are neither known as catalytic sites nor redox-sensitive, they might be modified through other routes than oxidation (e.g., S-palmitoylation, lipid electrophile-based adductions), or function in a non-catalytic manner. Those cysteines might still have be important and modulate protein activities.

In addition, it is important to note that the R_{100:10} value for each cysteine is regarded as a predictive rather than definitive indicator/parameter. Thus, the cutoff ratio values (0-3, 3-6, >6) were empirically determined to categorize all cysteines identified, and we adopted a standard commonly used in the field for the sake of simplicity and consistency (Weerapana et al., *Nature*, 2010; Martell et al., *Cell Chem Biol*, 2016; Petrova et al, *PNAS*, 2018).

In summary, to globally identify redox-regulated cysteines using *C.elegans* as a model, we quantitatively examined cysteines in its proteome from two distinct dimensions. First, we quantified the intrinsic reactivity of thousands of cysteines, which greatly expanded the landscape of the reactive cysteine proteome in *C. elegans* compared to previous studies (Kumsta, C. et al. *Antioxid Redox Signal*, 2011; Knoefler, D. et al. *Mol Cell*, 2012; Martell, J. et al. *Cell Chem Biol*, 2016). Our intrinsic reactivity dataset provides valuable prediction of a cysteine’s functionality within a broad range, but does not directly show its potential for redox regulation. Next, we extensively profiled three major redox forms (-SH, -SOH and -SO₂H) to directly assess peroxide-sensitive redox events in *C.elegans*. A total of 5453 -SH, 1521 -SOH, and 82 -SO₂H sites were identified and quantified, including many cysteines that were found for the first time as redox-regulated. Although each approach has its own limitations, our results dramatically increase the scale of *C.elegans* redoxome data, and provide a useful resource for the redox field.

[2] The authors have detected cysteines in various components of the MAPK pathway that are sensitive to oxidation. To test whether this oxidation might be important for the function of 2 of these kinases (MAPKK, SEK-1 and MAPK, PMK-1), they have generated *C. elegans* expressing Cys to Ser mutant versions of each. Analysis of these strains suggests the oxidation-sensitive cysteines are important for the phosphorylation of PMK-1. Indeed, assuming that SEK-1 mutant protein levels are unaffected (which should be shown), then their data provide strong evidence that C213 is important for the kinase activity and biological function of SEK-1 in protecting against infection (Fig. 5d, f and h).

Response: We were also interested in whether SEK-1 protein levels change in the cysteine-to-serine mutant strain. Because no commercial antibodies against the *C. elegans* SEK-1 protein are available, we purchased a SEK1/MKK4 antibody from Cell Signaling Technology (#9152), which was raised against the human SEK1 protein. Unfortunately, the antibody did not detect any specific bands of the expected size in wild-type animals that were absent in *sek-1* null animals. Because the antibody does not recognize the *C. elegans* SEK-1 protein, we cannot conclude whether or not the cysteine-to-serine mutation affects SEK-1 protein abundance. We have edited the Discussion of the manuscript to reflect this possibility (Page 15, line 2).

However, the presented data do not support the conclusion that oxidation of either cysteine is responsible for the increased PMK-1P in response to peroxide: The largest effect of each cysteine substitution (Fig.5d-g) seems to be on the levels of PMK-1P under physiological conditions, with H₂O₂ and tBOOH both inducing increased PMK-1P in animals bearing either the *sek-1* or *pmk-1* mutant alleles. This suggests that the reduced form of these cysteines are important for PMK-1 phosphorylation.

Response: As the reviewer pointed out, the redox-inert cysteine-to-serine mutations have a large effect on PMK-1 phosphorylation levels under physiological conditions, which suggests that basal levels of oxidation at those cysteines are important for PMK-1 phosphorylation. Although oxidants seemed to induce p38 phosphorylation in either mutant, such increase was not statistically significant (Fig. 5d-5e, Supplementary Fig. 6b). Furthermore, we have added new data showing that SKN-1 transcriptional activity that depends on PMK-1 phosphorylation did not seem to increase in response to t-BOOH in either mutant, as shown by mRNA levels of the SKN-1 target *gst-10* (Fig. 5h). By contrast, PMK-1 phosphorylation levels in the presence of oxidants between wild-type and mutant animals are significantly different (Fig. 5d-5e, Supplementary Fig. 6b), suggesting that those cysteines are critical for p38 activation by ROS. We have added the results of statistical analyses to the figures and clarified our reasoning in the Results section of the revised manuscript (Page 11-12).

Moreover, animals expressing the PMK-1 C173S mutant seem to have no reduction in their immunity to PA14. This suggests that, although required for PMK-1 phosphorylation, this cysteine is not important for PMK-1's innate immune function.

Response: As the reviewer suggested, it is possible that C173 in PMK-1 is dispensable for pathogen resistance. However, it is also likely that the mutation caused a partial loss of PMK-1 activity, but the effects were not large enough to be picked up by this particular assay. In other words, because the effect of *pmk-1* null mutation was small compared to that of *sek-1* null mutation (Fig. 5g), our assay may not be sensitive enough to detect any changes that were even smaller due to partial loss of function in PMK-1. The less severe phenotype of *pmk-1* null(*km25*) than *sek-1* null(*km4*) is not surprising, as previous studies

have shown that the PMK-2 protein functions redundantly with PMK-1 in at least the nervous system to regulate behavioral responses to pathogenic bacteria in *C. elegans* (Pagano et al., 2015). We cannot definitely conclude the need for C173 in pathogen response, and we have edited the manuscript to acknowledge the uncertainty but also explain our reasoning more clearly (Page 12, line 25-26).

[3] p. 3 The authors' proteomic studies were carried out with 5mM H₂O₂. Prolonged exposure to 5-10mM H₂O₂ is lethal to *C. elegans* (Kumsta et al 2011, Olahova et al 2008) so it seems misleading to describe this as 'low dose' H₂O₂. Other experiments were carried out with lower levels (1mM) so some explanation of the choice of concentrations for different experiments would be helpful?

Response: We thank the reviewer for pointing this out. Although prolonged exposure to peroxide affects *C. elegans* lifespan, a short-term H₂O₂ treatment has minimal effects on lifespan and behavior in *C. elegans* (Kumsta et al. *Antioxid Redox Signal*, 2011). Nevertheless, we agree with the reviewer that 5 mM H₂O₂ is better considered as a "modest" than "low" dose, and we have changed the statement in the manuscript.

The explanation for different doses used in different experiments is as follows: For the proteomic analysis, our goal was to detect early redox events or the most oxidation-sensitive cysteine residues. An acute treatment of 5 mM H₂O₂ for 5 min rapidly induces oxidation events at the most reactive cysteines, without harming the animals (Olahova, M. et al. *PNAS*, 2008) or causing secondary effects on protein synthesis or degradation that might affect protein abundance.

For all functional assays, we aimed to evaluate the role for cysteine oxidation under more physiological conditions, and a lower dose of oxidants was used to prevent animals from being over-stressed. Thus, in all experiments other than the proteomic analysis, 1mM H₂O₂ or 1 mM t-BOOH was used. In addition, biological events such as changes in mRNA levels and subcellular localization take longer to occur than the cysteine modification events themselves. For example, the effect of oxidants on PMK-1 phosphorylation is minimal after 15 min oxidant treatment but becomes robustly measurable after 30 min (Hourihan et al., *Mol Cell*, 2016). Therefore, worms were treated for 30 min or longer before measuring protein phosphorylation levels, mRNA levels, or subcellular localization. We have also clarified the "Stress treatment" part in the Methods (Page 21, line 27).

[4] Although the authors used H₂O₂ in their MS studies, in some of their subsequent expts, eg. (Fig.4) DAF-16 localisation and the effect of GCN-1 on p-eIF2a, they use tBOOH rather than H₂O₂? Some explanation for the altered choice of peroxide should be included or these experiments repeated with H₂O₂.

Response: In order to rule out any possible effects of fasting on worms and focus on the effects of oxidants, worms were fed with bacteria as food in all functional experiments.

The regular *E. coli* strain used to feed *C. elegans* expresses catalases/reductases that easily degrade H_2O_2 , but not t-BOOH (Schiffer *et al.*, *Elife*, 2020). Besides, t-BOOH is an organic peroxide that is more stable than H_2O_2 , and it has been frequently used in our lab for the sake of convenience and reproducibility. Thus, we initially performed all functional assays in Figures 4 and 5 with t-BOOH. To confirm that the phenotypes are general responses to ROS but not specific to t-BOOH, in our initial manuscript, we also tested the effects of H_2O_2 on p38 phosphorylation in the presence of a mutant *E. coli* strain that cannot degrade H_2O_2 (Seaver *et al.*, *J Bacteriol*, 2001). The use of H_2O_2 did not change our conclusion (Supplementary Fig. 4). To address the reviewer's concerns, we have now tested the effects of H_2O_2 on DAF-16 localization and p-eIF2a levels as well. These H_2O_2 results did not change our conclusion (**Figure R1**), and we have added them in the revised manuscript as Supplementary Fig. 4. We thank the reviewer for pointing out this concern and think that the paper has been improved by alleviating this uncertainty.

Figure R1/Supplementary Fig. 4. H_2O_2 treatment induces DAF-16::GFP nuclear accumulation and GCN-1-dependent eIF2 α phosphorylation. (a) Fluorescent images and quantification (b) showing nuclear localization of DAF-16::GFP with or without 1mM H_2O_2 (mean \pm SEM, n=3 experiments, at least 117 worms per condition). *, $P \leq 0.05$; Two-tailed Student's t-test. (c) Representative Western blots and quantification (d) showing eIF2 α phosphorylation levels in response to 1mM H_2O_2 treatment in wild-type and *gcn-1(-)* animals (mean \pm SEM, n=2 experiments). *, $P \leq 0.05$; ns, not significant; One-way ANOVA with Bonferroni post-test.

[5] Line 141 It is well established that the structural environment will influence the sensitivity of particular cysteines to oxidation. Therefore, it is not clear to me why the authors consider it intriguing that different cysteines in the same protein showed variable reactivity?

Response: The reviewer is right. It is not surprising to find that different cysteines in the same protein showed variable reactivity. The sentence therefore has been removed from the manuscript.

[6] line148 -The authors follow up describing how the active site cysteine is less reactive in one of the 4 GAPDH parologs with ‘we also noticed that the inherent reactivity of certain cysteines remained relatively constant through evolution’. Some explanation of this apparent contradiction is needed: Perhaps the point they are making is that their analysis does not reveal general rules, but examples of both conserved and non-conserved reactivity?

Response: We thank the reviewer for helping us clarify this point. To investigate the correlation between intrinsic reactivity of cysteines and their sequence-level conservation, we additionally compared the data set of the *C.elegans* cysteinome we obtained with what was previously generated from *homo sapiens* (Weerapana, E. *et al. Nature*, 2010). Notably, we found that more than half of conserved cysteines measured in both studies showed similar reactivity across evolution (**Figure R2/Supplementary Fig. 2**). In this regard, to avoid misunderstanding, we have rewritten the corresponding paragraph as follows:

“Our cysteine reactivity data set included many conserved cysteines for which reactivity was assessed previously in cultured human cells²². Interestingly, 54% of these cysteines were detected as reactive ($R_{100:10} \leq 6$) in both species (Supplementary Fig. 2). For example, the active cysteine C33 in the glutathione S-transferase GSTO-1 exhibited almost the same reactivity in *C. elegans* ($R_{100:10}=1.1$) (Fig. 2f) and *H. sapiens* ($R_{100:10}=0.9$)²². Conserved cysteines within parologs often but not always display similar $R_{100:10}$ values. For instance, C158 is an active site within all four orthologs (GPD-1 through GPD-4) of human glyceraldehyde 3-phosphate dehydrogenase (GAPDH)²⁷. C158 in GPD-1 and GPD-2/3 showed very similar $R_{100:10}$ values of 5.89 and 5.27, though C158 of GPD-4 exhibited a much higher $R_{100:10}$ of 10.0 (Fig. 2g). Those results suggest that conservation at the level of amino acid sequence may generally correlate with cysteine intrinsic reactivity, although reactivity could be influenced by subtle changes in flanking sequences.

In conclusion, by identifying 3560 intrinsically reactive cysteines ($R_{100:10} \leq 6$), our findings greatly expanded the landscape of the reactive cysteine proteome in *C. elegans*. Our analysis revealed that dramatic differences in reactivity may exist for different cysteines within the same protein, and that conservation of cysteine residues to an extent

predicts their redox-reactivity and possible redox regulation of their functionality.”

Figure R2/Supplementary Fig. 2. Pie chart showing the classification of conserved cysteines of which intrinsic reactivity was detected in both *C.elegans* and *H.sapiens*. Those conserved cysteines with similar reactivity in both species of are shown in dark red (with a fold-change between $R_{100:10}^{C.elegans}$ and $R_{100:10}^{H.sapiens}$ less than 1.5). Conserved cysteines exhibiting higher reactivity in *C.elegans* are shown in red, and those exhibiting higher reactivity in *H.sapiens* in light red. An intrinsic cysteine reactivity dataset of *H.sapiens* was retrieved from a previous report (Weerapana, E. et al. Quantitative reactivity profiling predicts functional cysteines in proteomes. *Nature*, 2010, 468: 790-795 doi:10.1038/nature09472).

Reviewer #2 (Remarks to the Author):

This manuscript describes the use of a suite of redox probes that assess cysteine SH, SOH and SO₂H in *C. elegans* treated with hydrogen peroxide to induce oxidative stress. The authors first identify hyperreactive cysteines in *C. elegans* lysates, and then proceed to quantifying changes in SH, SOH and SO₂H upon peroxide treatment. In general, the data show that SO₂H forms show the most dramatic increases upon peroxide treatment. The authors present a compendium of redox sensitive cysteines in *C. elegans* and identify pathways that are likely to be redox regulated. In particular, they focus on the MAPK signaling pathway and identify two cysteines on PMK1 and SEK1 that regulated downstream p38 activation.

This study provides a useful database of redox sensitive cysteines in *C. elegans* as a resource to the redox biology community. I recommend publication upon addressing the

comments below:

Response: We thank the reviewer for the positive comments about our work and its importance.

Figure R3/Supplementary Fig. 1. Reproducibility of quantitative chemoproteomics. (a) Violin plots of CV values of all the ratio values for quantifiable cysteines from at least two biological replicates. (b) Statistics defining Violin plots shown in (a).

1. For the mass spectrometry data provided, it is unclear how (or if) the data were filtered to obtain the final list. For example, the data should be filtered to only include those hits that provide ratios in 2 of 2, or 2 of 3 replicates, with hits that show high standard deviation removed. This would provide a more high confidence dataset for the community than the current data, which provides hits based on an effective n=1 ratio value.

Response: We thank the reviewer for raising this important issue about data processing. For the cysteine intrinsic reactivity data set, 2735 cysteines were quantified in at least two of the three biological replicates (Supplementary Data 1). For the redoxome data set,

3301 cysteine sites were measured repeatedly in at least two biological replicates (Supplementary Data 2-4). The overall quality and reproducibility of the quantification results was further assessed by plotting the distribution of coefficient of variation (CV) values of all repeatedly measured sites for each dataset. As shown in **Figure R3** (Supplementary Fig. 1), median CV values were lower or close to 20% for the determined ratio values obtained from two or three biological replicates, demonstrating the high reproducibility of the quantification results. We are therefore confident that the cysteine reactivity and H₂O₂-induced differences were both quantified in a trustworthy manner.

As the reviewer noticed, many cysteines were detected in only one replicate, which could be attributed to the stochastic nature of shotgun proteomics. In fact, many “single hits” were also active or redox-sensitive. One of such examples is a *bona fide* redox-sensitive cysteine (C52, $R_{T/C}^{IPM} = 0.46$, Supplementary Data 2) in the CXXC motif of protein disulfide-isomerase 1 (PDI-1). In addition, although C173 in PMK-1 was only detected in one replicate in our –SH dataset (Supplementary Data 2), this novel peroxide-sensitive site ($R_{T/C}^{IPM} = 0.57$) was then validated to be functionally important in our biological assays (Fig. 5 and Supplementary Fig. 6). Because the quantification results obtained from only one replicate might still be informative, filtering out those ‘single hits’ may result in the omission of potentially important sites. This phenomenon has also been noticed in previous analyses of the cysteine proteomes (Deng X et al., *Cell Host Microbe*, 2013; Martell et al., *Cell Chem Biol*, 2016; Petrova B et al., *PNAS*, 2018; Huang J et al., *PNAS*, 2019). With precautions being taken for the ‘single hits’, the current dataset is useful for discovering functionally important and physiologically relevant redox events in *C.elegans*. Therefore, we have decided to keep our original dataset in the Supplementary data. The reviewer has made an important point, however, in order to avoid misleading the readers, the aforementioned points have been incorporated into the Results section of the revised manuscript (Page 5, line 20-22; Page 7, line 23-32).

2. Figure 4f and g, what concentrations and times were used for the tBOOH treatment? Do these conditions mimic that used for the initial proteomics studies i.e. 5mM H₂O₂ for 5 mins?

Response: 1mM t-BOOH for 30 min was used in Fig. 4f-g. For clarity, we have added such information in the corresponding figure legend in addition to the Methods section. This is different from the conditions used in the proteomic studies, and we have explained the reasons for these differences in our response to questions 3 and 4 by reviewer #1.

3. Figure 3g, it is unclear if there is a clear bias for oxidation of odd numbered cysteines given the data provided. What is the percentage of even vs. odd cysteines for the entire proteome? Is this ratio skewed when looking at the pool of oxidized cysteines? In general, I felt the data does not strongly support this conclusion, and recommend removal of this section if the proteome-wide versus oxidized-cysteine ratios for odd versus even do not

change significantly.

Response: As suggested by the reviewer, Fig. 3g-h and its relevant description have been removed.

4. Figure 5h, cysteine mutation in *pmk-1* has no effect on survival, whereas the *pmk-1* null mutants show a distinct decrease in survival. The authors claim that this is due to functional redundancy with *pmk-2*. However, the presented data also supports the conclusion that this cysteine is not important for *pmk-1* function. A stronger case needs to be made to convince the readers that the lack of change is due to *pmk-2* redundancy instead of the cysteine just not being functionally relevant.

Response: Our Western blots and new qRT-PCR results indicate that C173 is important for PMK-1 function in p38 phosphorylation and activity (Fig. 5e-f, Supplementary Fig. 6b). Animals carrying the cysteine-to-serine mutation showed lower p38 phosphorylation levels and impaired PMK-1-dependent SKN-1 activity compared to wild-type animals, suggesting that the mutation leads to a partial loss-of-function. We didn't intend to claim that redundancy with PMK-2 is the only possibility for the lack of defects in pathogen resistance of this *pmk-1* mutation, although we believe that it is very likely to play a role. We have discussed this in our response to question 2 by reviewer #1, and have modified the text accordingly (Page 12, line 25-26).

Reviewer #3 (Remarks to the Author):

In this manuscript, Meng et al. characterize the redoxome of *C. elegans* by applying a redox-proteomic approach that enables the measurement of changes in three major cysteine redox forms after hydrogen peroxide treatment. The authors are able to assess the redox state of ~5000 cysteines, a several fold improvement compared to previous studies. They identify novel redox-sensitive cysteine residues, and generate redox-inactive worm mutants for some of these new candidates.

Overall, the manuscript is clearly written and logically structured, and the Figures are well presented. The study represents a valuable advance to the redox field, as well as appealing to broader audiences interested in proteomics, post-translational modifications, oxidative stress responses and cell signaling pathways. The full redoxome dataset generated will be an important resource for future studies and discoveries.

The following minor comments should be addressed before publication:

Response: We very much appreciate the reviewer's positive comments regarding the major points we made in the manuscript.

Figure 4

1) The authors find that GCN1 is widely oxidized in response to H₂O₂ treatment. In panels 4j/k, they show that oxidative stress by t-BOOH does not result in enhanced phosphorylation of eIF2a in GCN-1-null worms as seen in WT controls. The authors suggest that oxidant-reactive cysteines in GCN-1 might be involved in the redox regulation of GCN-2 activity. In this context, it would be helpful to know whether the levels of GCN-1 are altered in response to t-BOOH in WT worms?

Response: It would be interesting to know whether GCN-1 protein levels change upon t-BOOH treatment. However, commercial antibodies that detect the human GCN1 protein are not likely to detect the worm GCN-1 protein, because the epitope sequence is not well conserved. Thus, we cannot determine whether the levels of GCN-1 are altered by t-BOOH. Nevertheless, the reviewer brought up an interesting possibility, and we have added it to the revised manuscript (Page 14, line 9-10).

Figure 5

2) In panels 5d-g, it may be useful to have 'C213S' and 'C173S' along with the sek-1(syb1398) and pmk-1(syb1415) labels to clearly indicate that these are redox knock-in mutants, as this information is in the main text but not the Figure legend text. The same point is valid for the survival curve in panel 5h, to distinguish more explicitly between the redox mutants and the null mutants.

Response: We agree with the reviewer that such information would be helpful for readers. Therefore, we have added information about mutant strains to the accompanying legend.

3) On page 10 of the manuscript: "Immunoblotting showed that a C173S mutation (pmk-1(syb1415)) did not affect PMK-1 protein abundance (Fig. 5e and 5g)." This refers to the bands labelled as p38 on the blots, which may lead to slight confusion for the reader. Also is there an accompanying actin or other loading control?

Response: We thank the reviewers for clarifying this uncertainty, and we have replaced "p38" with "PMK-1" in the labels of those blots in Fig. 5.

As for the loading control, we measured protein concentrations of each sample using the BCA assay and loaded the same amount of proteins to each lane to ensure equal loading. This information has been added to the Methods section. When generating Western blots in Fig. 5, we only detected total and phospho- PMK-1, because the ratio of phosphorylated forms vs total protein levels indicates protein activity. However, we do understand the concern of the reviewer. To analyze whether or not the cysteine-to-serine mutation affects PMK-1 protein levels, we have conducted additional Western blots and measured total PMK-1 levels and actin levels in N2 (wild-type) and PMK-1 C173S mutants (Supplementary Fig. 6c). Our blots show that PMK-1 protein levels do not change in the mutant strain.

4) In panels 5d/f, the authors show that the SEK-1 C213S mutant results in diminished phosphorylation of p38 compared to its WT counterpart in response to t-BOOH and H₂O₂. However, the phospho-p38/p38 levels are also lower in the redox-inert mutant even under untreated conditions. Does this mean that C213 is not only necessary for modulation of SEK-1 activity in response to oxidative stress but also for its general activity?

Response: We thank the reviewer for bringing up this important point, and we agree with the reviewer that C213 is important for SEK-1 activity under both physiological conditions as well as oxidizing conditions. We have clarified it in the manuscript (Page 11, line 23-24).

The SEK-1 C213S mutant still seems to be responding to oxidative stress at the level of p38 phosphorylation i.e. the phospho-p38/p38 response ratio upon oxidative treatment relative to the respective untreated controls appears to be similar-is this the case?

Response: To address the question, we have conducted statistical analyses, and we have found that the oxidant-induced increase is not statistically significant in the mutant strain (Fig. 5d, Supplementary Fig. 6a). So, while we cannot definitively say that there is no response, it is certainly reduced. Furthermore, we have added new data showing that SKN-1 transcriptional activity that depends on PMK-1 phosphorylation did not seem to increase in response to t-BOOH in the SEK-1 mutant animals, as shown by mRNA levels of the SKN-1 target *gst-10* (Fig. 5h). We have also added the statistical analyses to Fig. 5 and Supplementary Fig. 6.

5) As shown by the authors for PMK-1, is SEK-1 protein stability/abundance affected by the C213S mutation?

Response: Unfortunately, this was not possible with existing reagents: please see our response to question 2 by reviewer #1.

Reviewer #1 (Remarks to the Author):

This paper presents a global analysis of cysteine reactivity that should be a valuable resource for identifying functional cysteines in *C. elegans* proteins, that may be regulated by redox changes. Indeed, the authors have identified reactive, oxidation-sensitive cysteines in a MAPKK(SEK-1) and a MAPK(PMK-1) that they show are important for the activity of these signaling proteins. The authors have addressed most of the concerns I had with their original submission. However, some minor issues with the interpretation/presentation of certain findings remain that should be addressed before publication. My main suggestion is that a sentence in the abstract should be edited, as the authors have not shown here that 'each component of the p38MAPK signaling pathway is redox-regulated' and it is unclear what they mean by 'in antioxidant response'. Instead, it seems more appropriate to state that they have identified oxidation-sensitive cysteines that are critical for the activity of these kinases.

Comments on revised manuscript and rebuttal:

Original point (1) regarding the absence of peroxiredoxins and other known reactive cysteines from their data sets: on p.8 the authors suggest a plausible technical explanation for why tryptic peptides containing peroxiredoxin cysteines might not have been detected by their LC-MS/MS. However, another explanation occurred to me: The peroxide-reacting cysteine of 2-Cys peroxiredoxins is so sensitive to oxidation that measures normally need to be taken to prevent its oxidation during protein extraction (Cox et al 2010). I appreciate that the authors added catalase to their samples to minimise post-lysis oxidation. However, is it possible that the peroxiredoxins are already in oxidised, disulphide-bonded forms in the native *C. elegans* extracts prior to labelling, preventing their reaction with IPM etc? This may not be the case, but is easy to check using immunoblotting, and would be consistent with data included in Hourihan et al 2016, that shows that the majority of PRDX-2 is in its disulfide-bonded form in *C. elegans* lysates prepared under native conditions. Inclusion of this additional experiment, would also be helpful because it would allow a direct comparison of how protein oxidation in their untreated samples relates to other studies which have employed different methodology. For instance, Knoepfler et al 2012 in which samples were prepared under acidic conditions to prevent post-lysis oxidation, and the ratio of PRDX-2 disulfides to reduced PRDX-2 used as a measure of physiological oxidation state. As the authors suggest on p.8 that 'the large number of sulfenylated proteins suggests that basal sulfenylation levels might be high in *C. elegans*' perhaps the authors might also appreciate this opportunity to strengthen this argument by eliminating the possibility that there is a greater level of basal cysteine oxidation with their approach.

Original point (2): The authors have demonstrated that the oxidation-sensitive cysteines they have identified in SEK-1 and PMK-1 are important for phosphorylation of PMK-1 under physiological conditions and, in the case of SEK-1, for its role in innate immunity. The authors newly added *gst-10* mRNA data is consistent with these cysteines being important for the response to peroxide. However, they have not established that the oxidation of either cysteine is important for activation of PMK-1 and innate immunity as their abstract suggests.

This is not to say that the discovery that these kinases have oxidation-sensitive cysteines that are important for their activity is not significant, and it certainly validates the capacity of their proteomic approach to discover functional cysteines. My concern is only that some of their conclusions wrt the effect of redox change/s in these cysteines on MAPKK and MAPK function seem premature based on the data they present. The authors may reasonably consider further analysis of why the identified cysteines in SEK-1 and PMK-1 are important for PMK-1 phosphorylation beyond the scope of their manuscript. However, based on the data they present, it seems more appropriate to simply conclude that these cysteines are important for PMK-1 phosphorylation. For example, their statement on p.11 that 'their data suggest that basal levels of cysteine oxidation at C213 is important for SEK-1 kinase' seems misleading given that C213 is more likely to be reduced under physiological conditions? Instead, I would suggest that their data indicate that a reduced, reactive cysteine thiol in this position may be important for SEK-1 activity and that a reduced reactive C173 is important for PMK-1 phosphorylation.

Other suggestions:

p.3 line 31 this sentence is not very clear. I suggest that 'ROS targets' should be ROS-targeted cysteines and 'their regulatory roles' should be replaced with 'the importance of these cysteines'

p.6 line 17: commas should be inserted around 'but not always'

Fig. 5F legend -should be C173S

Reviewer #2 (Remarks to the Author):

The revised manuscript addresses my previous concerns. I recommend publication.

Reviewer #3 (Remarks to the Author):

The manuscript by Meng and colleagues has been much improved upon revision. I believe the study will be a valuable resource for the redox field. My queries have been addressed satisfactorily by the authors, and I am pleased to recommend publication in Nature Communications.

REVIEWER COMMENTS

Reviewer #1 (Remarks to the Author):

This paper presents a global analysis of cysteine reactivity that should be a valuable resource for identifying functional cysteines in *C. elegans* proteins, that may be regulated by redox changes. Indeed, the authors have identified reactive, oxidation-sensitive cysteines in a MAPKK(SEK-1) and a MAPK(PMK-1) that they show are important for the activity of these signaling proteins. The authors have addressed most of the concerns I had with their original submission. However, some minor issues with the interpretation/presentation of certain findings remain that should be addressed before publication. My main suggestion is that a sentence in the abstract should be edited, as the authors have not shown here that 'each component of the p38MAPK signaling pathway is redox-regulated' and it is unclear what they mean by 'in antioxidant response'. Instead, it seems more appropriate to state that they have identified oxidation-sensitive cysteines that are critical for the activity of these kinases.

Response: We are glad that most previous concerns raised by the reviewer have been addressed, and appreciate the positive comments. We have edited the abstract. Specifically, the sentence "... determined that each p38 MAP kinase (MAPK) signaling pathway component is redox-regulated, and identified evolutionarily-conserved cysteines that are critical for p38 activation in antioxidant response and innate immunity." has been changed to "... identified redox-sensitive cysteines that are important for signaling through the p38 MAP kinase (MAPK) pathway". We have also clarified the idea of "antioxidant response (which is mediated by SKN-1/NRF2, the downstream effector of the p38MAPK signaling pathway)" within the results section (Page 11, line 28).

Comments on revised manuscript and rebuttal:

Original point (1) regarding the absence of peroxiredoxins and other known reactive cysteines from their data sets: on p.8 the authors suggest a plausible technical explanation for why tryptic peptides containing peroxiredoxin cysteines might not have been detected by their LC-MS/MS. However, another explanation occurred to me: The peroxide-reacting cysteine of 2-Cys peroxiredoxins is so sensitive to oxidation that measures normally need to be taken to prevent its oxidation during protein extraction (Cox et al 2010). I appreciate that the authors added catalase to their samples to minimise post-lysis oxidation. However, is it possible that the peroxiredoxins are already in oxidised, disulphide-bonded forms in the native *C. elegans* extracts prior to labelling, preventing their reaction with IPM etc? This may not be the case, but is easy to check using immunoblotting, and would be consistent with data included in Hourihan et al 2016, that shows that the majority of PRDX-2 is in its disulfide-bonded form in *C. elegans* lysates prepared under native conditions. Inclusion of this additional experiment, would

also be helpful because it would allow a direct comparison of how protein oxidation in their untreated samples relates to other studies which have employed different methodology. For instance, Knoepfler et al 2012 in which samples were prepared under acidic conditions to prevent post-lysis oxidation, and the ratio of PRDX-2 disulfides to reduced PRDX-2 used as a measure of physiological oxidation state. As the authors suggest on p.8 that ‘the large number of sulfenylated proteins suggests that basal sulfenylation levels might be high in *C. elegans*’ perhaps the authors might also appreciate this opportunity to strengthen this argument by eliminating the possibility that there is a greater level of basal cysteine oxidation with their approach.

Response: We thank the reviewer for raising this possibility, which we should have addressed in the manuscript. As the reviewer noted, it has been reported that in *C. elegans* lysates the hyperreactive cysteines in 2-cys peroxiredoxins (e.g., PRDX-2) are largely oxidized to disulfides under similar conditions (Hourihan et al *Mol Cell*, 2016). While strong acid conditions (e.g. Trichloroacetic acid used in Knoepfler et al 2012) can minimize post-lysis oxidation, in the QTRP analysis, we used non-denaturing buffer (pH 7.4, without ionic detergents) to maintain the intrinsic reactivity of profiled cysteines (Fu et al., *Nat Protoc*, 2020), with catalase added in the lysis buffer to prevent oxidation by a trace amount of peroxide generated during protein extraction. As the reviewer suggested, we examined whether PRDX-2 (as a representative of peroxiredoxin family) existed mainly in the disulfide-bonded form (lacking ability to react with the thiol-reactive IPM probe) in lysates obtained using our protocol. Our immunoblot shows that in *C. elegans* lysates the cysteines in PRDX-2 are indeed largely oxidized to disulfides (Fig. R1/Supplementary Fig. 3). Such result is consistent with what was published before, and we have added the possibility raised by this astute reviewer in the revised manuscript. Nonetheless, it is important to keep in mind that any post-lysis oxidation would have had the same effects on the H₂O₂-treated and control samples, and thus it would not have substantially affected our H₂O₂-dependent redoxome dataset and conclusions.

Fig. R1: Immunoblots of PRDX-2 and tubulin without and with 1 mM H₂O₂ treatment for 5 min, showing that the reactive cysteines in the peroxiredoxin PRDX-2 were predominantly disulfide-linked under both conditions, which is consistent with previous reports (Hourihan et al *Mol Cell*, 2016).

Original point (2): The authors have demonstrated that the oxidation-sensitive cysteines they have identified in SEK-1 and PMK-1 are important for phosphorylation of PMK-1 under physiological conditions and, in the case of SEK-1, for its role in innate immunity. The authors newly added *gst-10* mRNA data is consistent with these cysteines being important for the response to peroxide. However, they have not established that the oxidation of either cysteine is important for activation of PMK-1 and innate immunity as their abstract suggests.

This is not to say that the discovery that these kinases have oxidation-sensitive cysteines that are important for their activity is not significant, and it certainly validates the capacity of their proteomic approach to discover functional cysteines. My concern is only that some of their conclusions wrt the effect of redox change/s in these cysteines on MAPKK and MAPK function seem premature based on the data they present. The authors may reasonably consider further analysis of why the identified cysteines in SEK-1 and PMK-1 are important for PMK-1 phosphorylation beyond the scope of their manuscript. However, based on the data they present, it seems more appropriate to simply conclude that these cysteines are important for PMK-1 phosphorylation. For example, their statement on p.11 that 'their data suggest that basal levels of cysteine oxidation at C213 is important for SEK-1 kinase' seems misleading given that C213 is more likely to be reduced under physiological conditions? Instead, I would suggest that their data indicate that a reduced, reactive cysteine thiol in this position may be important for SEK-1 activity and that a reduced reactive C173 is important for PMK-1 phosphorylation.

Response: We thank the reviewer for helping us clarify our conclusion. We have therefore modified our language in the abstract (see above) as well as the result section (detailed as follow). Indeed, we demonstrate that SEK-1^{C213} and PMK-1^{C173} are both critical for basal activity of p38/PMK-1. Although the current data does not show whether the two cysteines exist mainly in a reduced or oxidized form under physiological conditions, it is important to note that this pathway becomes highly activated under oxidizing conditions (ROS or pathogen exposure), making it more likely that oxidation of these peroxide-sensitive cysteines regulates the activation of p38 (or phosphorylation of PMK-1).

The sentence "..., suggesting that basal levels of cysteine oxidation at C213 is important for SEK-1 kinase activity" has been changed to "..., indicating that the oxidation-sensitive cysteine C213 is important for the basal level of SEK-1 kinase activity".

The sentence "Together, these observations indicate that C213S is critical for SEK-1 activity in general" has been changed to "Together, these observations indicate that the

redox-sensitive cysteine C213 is critical for SEK-1 activity".

The sentence "Together, the data suggest that redox-reactive cysteines in the p38 pathway are critical for basal p38 activity, and that the activation of the p38 response to ROS depends on redox-reactive cysteines within proteins throughout the pathway" has been changed to "Together, the data suggest that redox-reactive cysteines in the p38 pathway are critical for basal p38 activity and p38 activation by ROS".

Other suggestions:

p.3 line 31 this sentence is not very clear. I suggest that 'ROS targets' should be ROS-targeted cysteines and 'their regulatory roles' should be replaced with 'the importance of these cysteines'

p.6 line 17: commas should be inserted around 'but not always'

Fig. 5F legend -should be C173S

Response: We thank the reviewer for these suggestions and have edited the text accordingly.

Reviewer #2 (Remarks to the Author):

The revised manuscript addresses my previous concerns. I recommend publication.

Response: We thank the reviewer for the valuable comments of our manuscript.

Reviewer #3 (Remarks to the Author):

The manuscript by Meng and colleagues has been much improved upon revision. I believe the study will be a valuable resource for the redox field. My queries have been addressed satisfactorily by the authors, and I am pleased to recommend publication in Nature Communications.

Response: We thank the reviewer for the valuable comments on the manuscript.